# SPARSE AUTOENCODERS REVEAL UNIVERSAL FEATURE SPACES ACROSS LARGE LANGUAGE MODELS

## ABSTRACT

We investigate feature universality in large language models (LLMs), a research field that aims to understand how different models similarly represent concepts in the latent spaces of their intermediate layers. Demonstrating feature universality allows discoveries about latent representations to generalize across several models. However, comparing features across LLMs is challenging due to polysemanticity, in which individual neurons often correspond to multiple features rather than distinct ones. This makes it difficult to disentangle and match features across different models. To address this issue, we employ a method known as *dictionary learning* by using sparse autoencoders (SAEs) to transform LLM activations into more interpretable spaces spanned by neurons corresponding to individual features. After matching feature neurons across models via activation correlation, we apply representational space similarity metrics like Singular Value Canonical Correlation Analysis to analyze these SAE features across different LLMs. Our experiments reveal significant similarities in SAE feature spaces across various LLMs, providing new evidence for feature universality.

## 1 INTRODUCTION

Large language models (LLMs) have demonstrated remarkable capabilities across a wide range of natural language tasks (Bubeck et al., 2023; Naveed et al., 2024; Liu et al., 2023). However, understanding how these models represent and process information internally remains a significant challenge (Bereska and Gavves, 2024). A key question in this domain is whether different LLMs learn similar internal representations, or if each model develops its own unique way of encoding linguistic knowledge. This question of feature universality is crucial for several reasons: it impacts the generalizability of interpretability findings across models, it could accelerate the development of more efficient training techniques, and it may provide insights into safer and more controllable AI systems (Chughtai et al., 2023; Gurnee et al., 2024; Sharkey et al., 2024; Bricken et al., 2023).

Comparing features across LLMs is inherently difficult due to the phenomenon of polysemanticity, where individual neurons often correspond to multiple features rather than distinct ones Elhage et al. (2022). This superposition of features makes it challenging to disentangle and match representations across different models. To address these challenges, we propose a novel approach leveraging sparse autoencoders (SAEs) to transform LLM activations into more interpretable spaces. SAEs allow us to decompose the complex, superposed representations within LLMs into distinct features that are easier to analyze and compare across models (Makhzani and Frey, 2013; Cunningham et al., 2023; Bricken et al., 2023). We then apply specific representational space similarity metrics to these SAE features, enabling a rigorous quantitative analysis of feature universality across different LLMs (Klabunde et al., 2023). Our methodology involves several key steps: obtaining SAEs trained on the activations of multiple LLMs to extract interpretable features, developing and applying representational similarity metrics tailored for high-dimensional SAE feature spaces, and conducting extensive experiments comparing SAE features across models with varying architectures, sizes, and training regimes.

To verify the effectiveness of our approach, we present a comprehensive set of experiments and results. Our experimental approach involves solving two issues for representational space similarity measurement: the permutation issue, which involves aligning individual neurons, and the rotational issue, which involves comparing two spaces which have different basis that are rotationally-invariant. Additionally, we present an approach to filter out feature matches with low semantically-meaningful

similarity. We demonstrate high correlations between individual SAE features extracted from distinct models, providing evidence for feature universality. Furthermore, we show that semantically meaningful subspaces (e.g., features related to specific concepts, such as Emotions) exhibit even higher similarity across models. We also analyze how feature similarity varies across different layers of LLMs, revealing patterns in how representations evolve through the model depth.

This work opens up several exciting avenues for future research. Exploring whether universal features can be leveraged to improve transfer learning or model compression techniques may lead to more efficient training paradigms (Yosinski et al., 2014; Hinton et al., 2015). Additionally, examining the implications of feature universality for AI safety and control, particularly in identifying and mitigating potentially harmful representations, could contribute to the development of safer AI systems (Hendrycks et al., 2023). By shedding light on the internal representations of LLMs, our research contributes to the broader goal of developing more interpretable, efficient, and reliable AI systems (Barez et al., 2023). Understanding the commonalities and differences in how various models encode information is a crucial step toward building a more comprehensive theory of how large language models process and generate human language. **Our key contributions include:**

1. A novel framework for comparing LLM representations using SAEs and similarity metrics.

2. Empirical evidence supporting the existence of universal features across diverse LLMs.

3. A quantitative analysis of how feature similarity varies across layers and semantic domains.

4. A step-by-step approach to compare feature weight representations by paired features that differs from previous methods which compare feature activations.

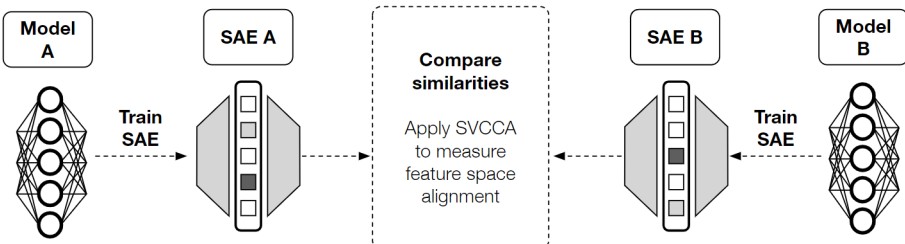

Figure 1: We train SAEs on LLMs, and then compare their SAE feature space similarity using metrics such as Singular Value Canonical Correlation Analysis (SVCCA).

## 2 BACKGROUND

**Sparse Autoencoders.** Sparse autoencoders (SAEs) are a type of autoencoder designed to learn efficient, sparse representations of input data by imposing a sparsity constraint on the hidden layer (Makhzani and Frey, 2013). The network takes in an input $\mathbf{x} \in \mathbb{R}^n$ and reconstructs it as an output $\hat{\mathbf{x}}$ using the equation $\hat{\mathbf{x}} = \mathbf{W}'\sigma(\mathbf{W}\mathbf{x} + \mathbf{b})$, where $\mathbf{W} \in \mathbb{R}^{h \times n}$ is the encoder weight matrix, $\mathbf{b}$ is a bias term, $\sigma$ is a nonlinear activation function, and $\mathbf{W}'$ is the decoder matrix, which often uses the transpose of the encoder weights. SAE training aims to both encourage sparsity in the activations $\mathbf{h} = \sigma(\mathbf{W}\mathbf{x} + \mathbf{b})$ and to minimize the reconstruction loss $\|\mathbf{x} - \hat{\mathbf{x}}\|_2^2$.

The sparsity constraint encourages a large number of the neuron activations to remain inactive (close to zero) for any given input, while a few neurons, called *feature neurons*, are highly active. Recent work has replaced the regularization term by using the TopK activation function as the nonlinear activation function, which selects only the top K feature neurons with the highest activations (Gao et al., 2024), zeroing out the other activations. The active feature neurons tend to activate only for specific concepts in the data, promoting monosemanticity. Therefore, since there is a mapping from LLM neurons to SAE feature neurons that "translates" LLM activations to features, this method is a type of *dictionary learning* (Olshausen and Field, 1997).

**SAE Feature Weight Spaces.** By analyzing UMAPs of feature weights from an SAE trained on a middle residual stream layer of Claude 3 Sonnet, researchers discovered SAE feature spaces organized in neighborhoods of semantic concepts, allowing them to identify subspaces corresponding

to concepts such as biology and conflict (Templeton et al., 2024). Additionally, these concepts appear to be organized in hierarchical clusters, such as "disease" clusters that contain sub-clusters of specific diseases like flus. In Section §4.3, we measure the extent of similarity of these semantic feature spaces across LLMs.

**Representational Space Similarity:** Representational similarity measures typically compare neural network activations by assessing the similarity between activations from a consistent set of inputs (Klabunde et al., 2023). In this paper, we take a different approach by comparing the representations using the decoder weight matrices $W'$ of SAEs, where the columns correspond to the feature neurons. We calculate a similarity score $m(W, W')$ for pairs of representations $W$ in SAE $A$ and $W'$ from SAE B. We obtain two scores via the following two representational similarity metrics:

**Singular Value Canonical Correlation Analysis (SVCCA).** Canonical Correlation Analysis (CCA) seeks to identify pairs of canonical variables, $\mathbf{u}_i$ and $\mathbf{v}_i$, from two sets of variables, $\mathbf{X} \in \mathbb{R}^{n \times d_1}$ and $\mathbf{Y} \in \mathbb{R}^{n \times d_2}$, that are maximally correlated with each other (Hotelling, 1936). Singular Value Canonical Correlation Analysis (SVCCA) (Raghu et al., 2017) enhances CCA by first applying Singular Value Decomposition (SVD) to both $\mathbf{X}$ and $\mathbf{Y}$, which can be expressed as:

$$\mathbf{X} = \mathbf{U}_X \mathbf{S}_X \mathbf{V}_X^T, \quad \mathbf{Y} = \mathbf{U}_Y \mathbf{S}_Y \mathbf{V}_Y^T$$

where $\mathbf{U}_X$ and $\mathbf{U}_Y$ are the matrices containing the left singular vectors (informative directions), and $\mathbf{S}_X$ and $\mathbf{S}_Y$ are diagonal matrices containing the singular values. By projecting the original data onto these informative directions, noise is reduced. SVCCA then applies CCA on the transformed datasets, $\mathbf{U}_X$ and $\mathbf{U}_Y$, resulting in correlation scores between the most informative components. These scores are averaged to obtain a similarity score.

**Representational Similarity Analysis (RSA).** Representational Similarity Analysis (RSA) (Kriegeskorte et al., 2008) works by first computing, for each space, a Representational Dissimilarity Matrix (RDM) $\mathbf{D} \in \mathbb{R}^{n \times n}$, where each element of the matrix represents the dissimilarity (or similarity) between every pair of data points within a space. The RDM essentially summarizes the pairwise distances between all possible data pairs in the feature space. A common distance metric used is the Euclidean distance. After an RDM is computed for each space, a correlation measurement like *Spearman's rank correlation coefficient* is applied to the pair of RDMs to obtain a similarity score. RSA has been applied to measure relational similarity of stimuli across brain regions and of activations across neural network layers (Klabunde et al., 2023).

**SVCCA vs RSA.** SVCCA focuses on the comparison of entire vector spaces by aligning the principal components (singular vectors) of two representation spaces and then computing the canonical correlations between them. In other words, it identifies directions in the space that carry the most variance and checks how well these align between two models or layers. On the other hand, RSA is primarily concerned with the structure of relationships, or *relational structure*, between different data points (like stimuli) within a representation space. It allows us to measure how well relations such as king-queen transfer across models. Thus, SVCCA focuses on comparing the aligned subspaces of two representation spaces, while RSA is more suitable for analyzing the structural similarity between representations at the level of pairwise relationships.

## 3 METHODOLOGY

Representational similarity is important for capturing the across-model generalizations of both: 1) Feature Spaces, which are spaces spanned by groups of features, and 2) Feature Relations. We compare SAEs trained on layer $A_i$ from LLM $A$ and layer $B_j$ from LLM $B$. This is done for every layer pair. To compare latent spaces using our selected metrics, we have to solve both permutation and rotational alignment issues. For the permutation issue, we have to find a way to pair neuron weights. Typically, these metrics are performed by pairing activation vectors from two models according to common input instances. For example, input $I_k$ is passed through models A and B, and then a vector of $N_1$ neuron activations from model $A$ is compared to a paired vector of $N_2$ activations from model $B$. The common input instance creates individual mappings from one space to another, which is required for metrics like SVCCA and RSA (Klabunde et al., 2023).

In our paper, we do not pair activation vectors by input instances, but we pair the feature neuron weights themselves. This means we have to find individual pairings of "similar" neurons in SAE $A$ to SAE $B$. However, we do not know which features map to which features; the order of these features are permuted in the weight matrices, and some features may not have a corresponding partner. Therefore, we have to pairwise match them via a correlation metric. For the rotational issue, models typically develop their own basis for latent space representations, so features may be similar relation-wise across spaces, but not rotation-wise due to differing basis. To solve this issue, we employ rotation-invariant metrics.

*Assessing Scores with Baseline.* We follow the approach of Kriegeskorte et al. (2008) to randomly pair features to obtain a baseline score. We compare the score of the features paired by correlation (which we refer to as the "paired features") with the average score of $N$ runs of randomly paired features to obtain a p-value score. If the p-value is less than 0.05, the metric suggests that the similarity is statistically meaningful.

In summary, the steps to carry out our similarity comparison experiments are given below, and steps 1 to 3 are illustrated in Figure 8:

1. For the permutation issue: Get mean activation correlation of two decoder SAEs for similar LLMs. Pair each feature A with its max activation correlated feature from model B.

2. Rearrange the order of features in each matrix to pair them row by row.

3. For the rotational issue: Apply various rotation-invariant metrics to get a *Paired Score*.

4. Using the same metrics, obtain the similarity scores of $N$ *random pairings* to estimate a null distribution. Obtain a p-value of where the paired score falls in the null distribution to determine statistical significance.

**Semantic Subspace Matching Experiments.** For these experiments, we first identify semantically similar subspaces, and then compare their similarity. This is summarized by the following steps:

1. We find a group of related features in each model by searching for features which activate highly on the top 5 samples' tokens that belong to a *concept category*. For example, the concept of "emotions" contains keywords such as "rage". Since the correlations are calculated by tokens, we use single-token keywords. If a keyword from a concept appears at least once, that feature is included. The keywords in each category are given in Appendix D.

2. We obtain mappings between these subsets of features using max activation correlation.

3. We characterize the relationships of features within these groups (eg. pairwise differences between features), and compare these relationships across models using similarity metrics.

*Assessing Scores with Baseline.* We use two types of tests which compare the paired score to a null distribution. Each test examines that the score is rare under a certain null distribution assumption:

1. Test 1: We compare the score of a semantic subspace mapping to the mean score of randomly shuffled pairings of the same subspaces. This test shows just comparing the subspace of features is not enough; the features must be paired.

2. Test 2: We compare the score of a semantic subspace mapping to a mean score of mappings between randomly selected feature subsets of the same sizes as the semantic subspaces. This uses a correlation matrix only between the two subsets, not between the entire space. This shows that the high similarity score does not hold for any two subspaces of the same size.

Metrics for the Permutation Issue: These metrics measure individual, local similarity.

**Mean Activation Correlation.** We take the mean activation correlation between neurons, following the approach of Bricken et al. (2023). We pass a dataset of samples through both models. Then, for each feature in model A, we find its activation correlation with every feature in model B, using tokens as the common instance pairing. For each feature in model A, we find its highest correlated feature in model B. We pair these features together when conducting our experiments. We refer to scores obtained by pairing via mean activation correlation as *Paired Scores*.

**Filter by Top Tokens.** We notice that there are many "non-concept mappings" with "non-concept features" that have top 5 token activations on end-of-text / padding tokens, spaces, new lines, and

punctuation; these non-concept mappings greatly bring our scores down as they are not accurately pairing features representing concepts, as we describe further in Appendix B. By filtering max activation correlation mappings to remove non-concept mapping features, we significantly raise the similarity scores. We also only keep mappings that share at least one keyword in their top 5 token activations. The list of non-concept keywords is given in the Appendix B.

**Filter Many-to-1 Mappings.** We find that some of these mappings are many-to-1, which means that more than one feature maps to the same feature. We found that removing many-to-1 mappings slightly increased the scores, and we discuss possible reasons why this may occur in Appendix A. However, the scores still show high similarity even with the inclusion of many-to-1 mappings, as seen in Figures 2. We show only scores of 1-to-1 mappings (filtering out the many-to-1 mappings) in the main text, and show scores of many-to-1 mappings in Appendix A. We use features from the SAEs of Pythia-70m and Gemma-1-2B as "many" features which map onto "one" feature in the SAEs of Pythia-160m and Gemma-2-2B.

**Metrics for the Rotational Issue**: Rotation-invariant metrics measure global similarity after feature pairing. We apply metrics which measure: 1) How well subspaces align using **SVCCA**, and 2) How similar feature relations like king-queen are using **RSA**. We set the inner similarity function using Pearson correlation the outer similarity function using Spearman correlation, and compute the RDM using Euclidean Distance.

# 4 EXPERIMENTS

## 4.1 EXPERIMENTAL SETUP

**LLM Models.** We compare LLMs that use the Transformer model architecture (Vaswani et al., 2017). We compare models that use the same tokenizer because the mean activation correlation pairing relies on comparing two activations using the same tokens. We compare the residual streams of Pythia-70m, which has 6 layers and 512 dimensions, vs Pythia-160m, which has 12 layers and 768 dimensions (Biderman et al., 2023). We compare the residual streams of Gemma-1-2B, which has 18 layers and 2048 dimensions, to Gemma-2-2B, which has 26 layers and 2304 dimensions Team et al. (2024a;b). [1]

**SAE Models.** For Pythia, we use pretrained SAEs with 32768 feature neurons trained on residual stream layers from Eleuther (EleutherAI, 2023). For Gemma, we use pretrained SAEs with 16384 feature neurons trained on residual stream layers (Lieberum et al., 2024) We access pretrained Gemma-1-2B SAEs for layers 0, 6, 10, 12, and 17, and Gemma-2-2B SAEs for layers 0 to 26, through the SAELens library (Bloom and Chanin, 2024).

**Datasets.** We obtain SAE activations using OpenWebText, a dataset that scraps internet content (Gokaslan and Cohen, 2019). We use 100 samples with a max sequence length of 300 for Pythia for a total of 30k tokens, and we use 150 samples with a max sequence length of 150 for Gemma for a total of 22.5k tokens. Top activating dataset tokens were also obtained using OpenWebText samples.

**Computing Resources.** We run experiments on an A100 GPU.

## 4.2 SAE SPACE SIMILARITY

In summary, for almost all layer pairs after layer 0, we find the p-value of SAE experiments for SVCCA, which measures rotationally-invariant alignment, to be low; all of them are between 0.00 to 0.01 using 100 samples.[2] While RSA also passes many p-value tests, their values are much lower, suggesting a relatively weaker similarity in terms of pairwise differences. It is possible that many features are not represented in just one layer; hence, a layer from one model can have similarities to multiple layers in another model. We also note that the first layer of every model, layer 0, has a very high p-value when compared to every other layer, including when comparing layer 0 from another model. This may be because layer 0 contains few discernible, meaningful, and comparable features.

---

[1] Gemma-2 introduces several architectural improvements over Gemma-1 (Team et al., 2024a;b).

[2] Only 100 samples were used as we found the variance to be low for the null distribution, and that the mean was similar for using 100 vs 1k samples.

As shown in Figure 9 for Pythia and Figure 10 for Gemma in Appendix E, we also find that mean activation correlation does not always correlate with the global similarity metric scores. For instance, a subset of feature pairings with a moderately high mean activation correlation (eg. 0.6) may yield a low SVCCA score (eg. 0.03).

Because LLMs, even with the same architecture, can learn many different features, we do not expect entire feature spaces to be highly similar; instead, we hypothesize there may be highly similar *subspaces*. In other words, there are many similar features universally found across SAEs, but not every feature learned by different SAEs is the same. Our experimental results support this hypothesis.

**Pythia-70m vs Pythia-160m .** In Figure 4, we compare Pythia-70m, which has 6 layers, to Pythia-160m, which has 12 layers. When compared to Figures 11 and 12 in Appendix E, for all layers pairs (excluding Layers 0), and for both SVCCA and RSA scores, almost all the p-values of the residual stream layer pairs are between 0% to 1%, indicating that the feature space similarities are statistically meaningful. In particular, Figure 4 shows that Layers 2 and 3, which are middle layer of Pythia-70m, are more similar by SVCCA and RSA scores to middle layers 4 to 7 of Pythia-160m compared to other layers. Figure 4 also shows that Layer 5, the last residual stream layer of Pythia-70m, is more similar to later layers of Pythia-160m than to other layers. The number of features after filtering non-concept features and many-to-1 features are given in Tables 5 and 6 in Appendix E.

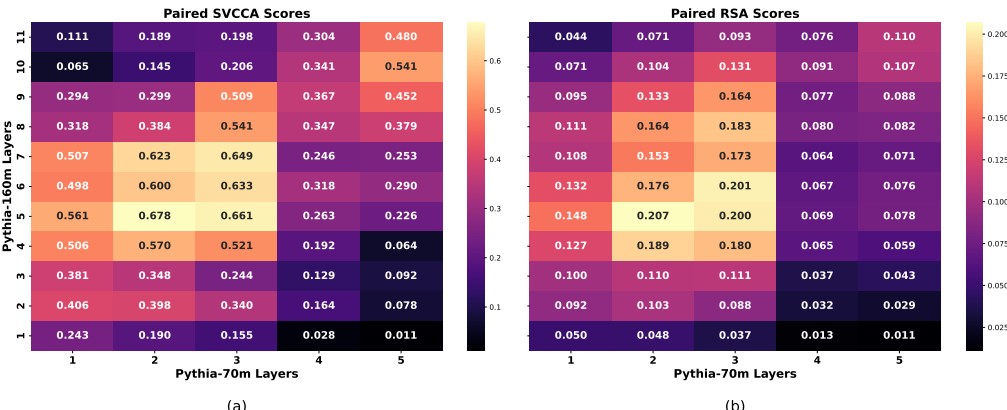

(a)                                            (b)

Figure 2: (a) SVCCA and (b) RSA 1-1 paired scores of SAEs for layers in Pythia-70m vs layers in Pythia-160m. We find that middle layers have the most similarity with one another (as shown by the high-similarity block spanned by layers 1 to 3 in Pythia-70m and Layers 4 to 7 in Pythia-160m). We exclude layers 0, as we observe they always have non-statistically significant similarity. The 1-1 scores are slightly higher for most of the Many-to-1 scores shown in Figure 4, and the SVCCA score at L2 vs L3 for 70m vs 160m is much higher.

**Gemma-1-2B vs Gemma-2-2B.** As shown by the paired SVCCA and RSA results for these models in Figure 6, we find that using SAEs, we can detect highly similar features in layers across Gemma-1-2B and Gemma-2-2B. Compared to mean random pairing scores and p-values in Figures 13 and 14 in Appendix E, for all layers pairs (excluding Layers 0), and for both SVCCA and RSA scores, almost all the p-values of the residual stream layer pairs are between 0% to 1%, indicating that the feature space similarities are statistically meaningful. The number of features after filtering non-concept features and many-to-1 features are given in Tables 7 and 8 in Appendix E.

Rather than just finding that middle layers are more similar to middle layers, we also find that they are the best at representing concepts. This is consistent with work by (Rimsky et al., 2024), which find the best steering vectors in middle layers for steering behaviors such as sycophancy. Other papers find that middle layers tend to provide better directions for eliminating refusal behavior (Arditi et al., 2024) and work well for representing goals and finding steering vectors in policy networks (Mini et al., 2023). In fact, we find that early layers do not represent universal features well, perhaps due to lacking content, while later layers do not represent them as well as middle layers, perhaps due to being too specific/complex.

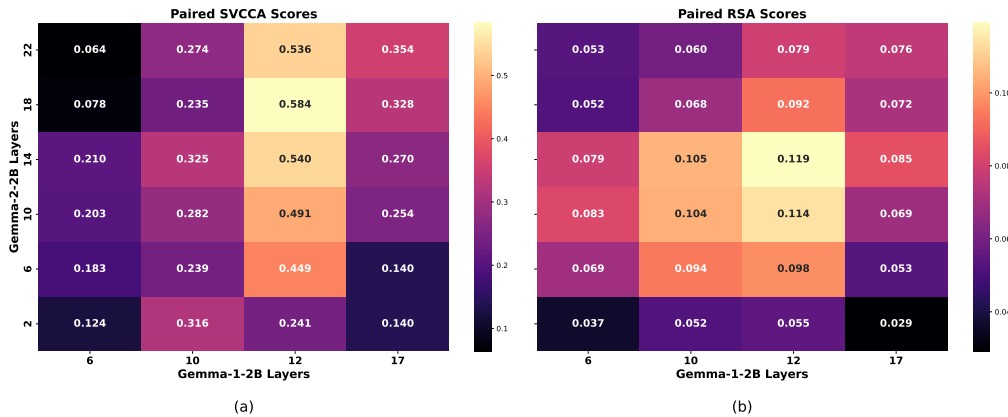

Figure 3: (a) SVCCA and (b) RSA 1-1 paired scores of SAEs for layers in Gemma-1-2B vs layers in Gemma-2-2B. Middle layers have the best performance. The later layer 17 in Gemma-1 is more similar to later layers in Gemma-2. Early layers like Layer 2 in Gemma-2 have very low similarity. We exclude layers 0, as we observe they always have non-statistically significant similarity. The 1-1 scores are slightly higher for most of the Many-to-1 scores shown in Figure 6.

Table 1: SVCCA scores and random mean results for 1-1 semantic subspaces of L3 of Pythia-70m vs L5 of Pythia-160m. P-values are taken for 1000 samples in the null distribution.

| Concept Subspace | Number of Features | Paired Mean | Random Shuffling Mean | p-value |
|---|---|---|---|---|
| Time | 228 | 0.59 | 0.05 | 0.00 |
| Calendar | 126 | 0.65 | 0.07 | 0.00 |
| Nature | 46 | 0.50 | 0.12 | 0.00 |
| Countries | 32 | 0.72 | 0.14 | 0.00 |
| People/Roles | 31 | 0.50 | 0.15 | 0.00 |
| Emotions | 24 | 0.83 | 0.15 | 0.00 |

## 4.3 SIMILARITY OF SEMANTICALLY MATCHED SAE FEATURE SPACES

We find that for all layers and for all concept categories, Test 2 described in §3 is passed. Thus, we only report specific results for Test 1 in Tables 1 and 2. Overall, in both Pythia and Gemma models and for many concept categories, we find that semantic subspaces are more similar to one another than non-semantic subspaces.

**Pythia-70m vs Pythia-160m.** We compare every layer of Pythia-70m to every layer of Pythia-160m for several concept categories. While many layer pairs have similar semantic subspaces. Middle layers appear to have the semantic subspaces with the highest similarity. Table 1 demonstrates one example of this by comparing the SVCCA score for layer 3 to layer 5 of Pythia-160m, which shows high similarity for semantic subspaces across models, as the concept categories all pass Test 1, having p-values below 0.05. We show the scores for other layer pairs and categories in Figures 15 and 17 in Appendix E, which include RSA scores in Table 9.

**Gemma-1-2B vs Gemma-2-2B.** In Table 2, we compare L12 of Gemma-1-2B vs L14 of Gemma-2-2B. As shown in Figure 6, this layer pair has a very high similarity for most concept spaces; as such, they likely have high similar semantically-meaningful feature subspaces. Notably, not all concept group are not highly similar; for instance, unlike in Pythia, the Country concept group does not pass the Test 1 as it has a p-value above 0.05. We show the scores for other layer pairs and categories in Figures 19 and 21 in Appendix E.

Table 2: SVCCA scores and random mean results for 1-1 semantic subspaces of L12 of Gemma-1-2B vs L14 of Gemma-2-2B. P-values are taken for 1000 samples in the null distribution.

| Concept Subspace | Number of Features | Paired SVCCA | Random Shuffling Mean | p-value |
|:---:|:---:|:---:|:---:|:---:|
| Time | 228 | 0.46 | 0.06 | 0.00 |
| Calendar | 105 | 0.54 | 0.07 | 0.00 |
| Nature | 51 | 0.30 | 0.11 | 0.01 |
| Countries | 21 | 0.39 | 0.17 | 0.07 |
| People/Roles | 36 | 0.66 | 0.15 | 0.00 |
| Emotions | 35 | 0.60 | 0.12 | 0.00 |

## 5 RELATED WORK

**Superposition and Sparse Autoencoders.** Superposition is a phenomenon in which a model, in response to the issue of having to represent more features than it has parameters, learns feature representations distributed across many parameters (Elhage et al., 2022). This causes its parameters, or neurons, to be polysemantic, which means that each neuron is involved in representing more than one feature. This leads to issues for interpretability, which aims to disentangle and cleanly identify features in models in order to manipulate them, such as by steering a model away from deception (Templeton et al., 2024). To address the issue of polysemanticity, Sparse Autoencoders (SAEs) have been applied to disentangle an LLM's polysemantic neuron activations into monosemantic "feature neurons", which are encouraged to represent an isolated concept (Makhzani and Frey, 2013; Cunningham et al., 2023; Gao et al., 2024; Rajamanoharan et al., 2024a;b). Features interactions in terms of circuits have also been studied (Marks et al., 2024).

**Feature Universality.** To the best of our knowledge, only Bricken et al. (2023) has done a quantitative study on individual SAE feature similarity for two 1-layer toy models, finding that individual SAE feature correlations are stronger than individual LLM neuron correlations; however, this study did not analyze the global properties of feature spaces. Gurnee et al. (2024) found evidence of universal neurons across language models by applying pairwise correlation metrics, and taxonimized families of neurons based on patterns of downstream, functional effects. O'Neill et al. (2024) discover "feature families", which represent related hierarchical concepts, in SAEs across two different datasets. Recent work has shown that as vision and language models are trained with more parameters and with better methods, their representational spaces converge to more similar representations (Huh et al., 2024).

**Representational Space Similarity.** Previous work has studied neuron activation spaces by utilizing metrics to compare the geometric similarities of representational spaces (Raghu et al., 2017; Wang et al., 2018; Kornblith et al., 2019; Klabunde et al., 2024; 2023; Kriegeskorte et al., 2008). It was found that even models with different architectures may share similar representation spaces, hinting at feature universality. However, these techniques have yet to be applied to the feature spaces of sparse autoencoders trained on LLMs. Moreover, these techniques compare the similarity of paired input activations. Our work differs as it compares the similarity of paired *feature weights*. Essentially, previous works do not show that one can match LLMs by features that both LLMs find in common.

**Feature Manifolds.** In real world data, features may lie on a manifold, where nearby features respond to similar data. (Olah and Batson, 2023; Bricken et al., 2023). Previous work has discovered the existence of features corresponding to months and days of the week arranged in a circular structure in SAE feature spaces across models (Engels et al., 2024). Studying how feature arrangements generalize across models may shed light on how features lie on a manifold.

**Mechanistic Interpretability.** Recent work has made notable progress in neuron interpretation (Foote et al., 2023; Garde et al., 2023) and interpreting the interactions between Attention Heads and MLPs (Neo et al., 2024). Other work in mechanistic interpretability has traditionally focused on 'circuits'-style analysis (Elhage et al., 2021), as well as the use of steering vectors to direct models at inference using identified representations (Zou et al., 2023; Turner et al., 2023; Bereska and Gavves, 2024). These approaches have been combined with SAEs to steer models in the more interpretable SAE feature space (Nanda and Conmy, 2024). Steering vectors have been found across models;

for instance, Arditi et al. (2024) found "Refusal Vectors" that steered a model to refuse harmful instructions across 13 different models, including Gemma and Llama-2.

**AI Safety.** Advanced AI systems may not be well aligned with the values of their designers (Ngo et al., 2022), due to phenomena such as goal misgeneralization (Shah et al., 2022; Langosco et al., 2022), deceptive alignment (Hubinger et al., 2024; 2021), sycophancy (Sharma et al., 2023), and reward tampering (Denison et al., 2024). SAEs may be able to find AI safety-related features across models; using SAEs to reliably find universal capabilities, especially in combination with control methods such as steering, may help provide solutions to the above pheonomena.

## 6 CONCLUSION

In this study, we address the underexplored domain of feature universality in sparse autoencoders (SAEs) trained on LLMs with multiple layers and across multiple model pairs. To achieve these insights, we developed a novel methodology to pair features with high mean activation correlations, and then assess their global similarity of the resulting subspaces using SVCCA and RSA. Our findings reveal a high degree of similarity in SAE feature spaces across various models. Furthermore, our research reveals that subspaces of features associated with specific semantic concepts, such as calendar or people tokens, demonstrate remarkably high similarity across different models. This suggests that certain semantic feature subspaces are universally encoded across varied LLM architectures. Our work enhances understanding of how LLMs represent information at a fundamental level, opening new avenues for further research into model interpretability and the transferability of learned features.

## LIMITATIONS

We perform analysis on available pre-trained SAEs for similar LLMs that use the same tokenizer, and preferrably on SAEs with the same number of features to match them. However, there are a limited number of pre-trained SAEs on large models. Our study is also limited by the inherent challenges in interpreting and comparing high-dimensional feature spaces. The methods we used to measure similarity, while robust, may not capture all nuances of feature interactions and representations within the SAEs. Additionally, our analysis is constrained by the computational resources available, which may have limited the scale and depth of our comparisons. The generalizability of our findings to SAEs trained on more diverse datasets or specialized domain-specific LLMs remains an open question. Furthermore, the static nature of our analysis does not account for potential temporal dynamics in feature representation that might occur during the training process of both LLMs and SAEs.

## FUTURE WORK

We performed similar analysis on the Pythia and Gemma LLMs that the SAEs in this paper were trained on, and found that they have some similarity, but much lower than the SAEs (around 0.1-0.2 for all layer pairs) with no patterns showing that the middle layers were the most similar; however, the LLMs did not have corresponding weights for the "residual stream", which is an addition of activations from the MLP and attention head layers, so we compared the MLPs of LLMs. Additionally, unlike the SAE dimensions, the LLM layers they were trained on had different dimensions (eg. 512 for Pythia-70m vs 768 Pythia-160m), making feature matching less trivial. Thus, future work could study the comparison of LLM similarity by feature weights to SAEs. Additional experiments could include using ground-truth feature experiments for LLMs that partially share ground-truth features to see if SAEs are modeling the same features, and performing functional similarity experiments.

## REPRODUCIBILITY

We include the code and instructions for reproducing our main results in the Supplementary Materials.

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

## A    NOISE OF MANY-TO-1 MAPPINGS

Overall, we do not aim for mappings that uniquely pair every feature of both spaces as we hypothesize that it is unlikely that all the features are the same in each SAE; rather, we are looking for large subspaces where the features match by activations and we can map one subspace to another.

We hypothesize that these many-to-1 features may contribute a lot of noise to feature space alignment similarity, as information from a larger space (eg. a ball) is collapsed into a dimension with much less information (eg. a point). Thus, when we only include 1-1 features in our similarity scores, we receive scores with much less noise.

As shown in the comparisons of Figure 4 to Figure 5, we find the 1-1 feature mappings give slightly higher scores for many layer pairs, like for the SVCCA scores at L2 vs L3 for Pythia-70m vs Pythia-160m, though they give slightly lower scores for a few layer pairs, like for the SVCCA scores at L5

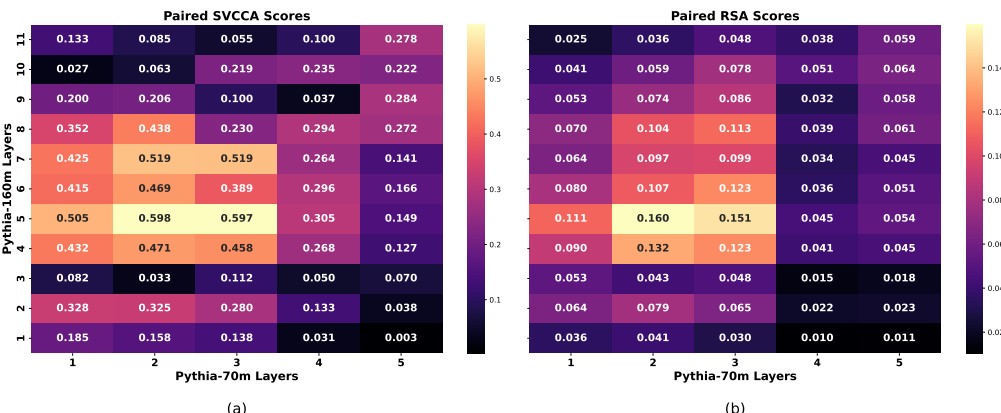

Figure 4: (a) SVCCA and (b) RSA **Many-to-1** paired scores of SAEs for layers in Pythia-70m vs layers in Pythia-160m. We note there appears to be an "anomaly" at L2 vs L3 with a low SVCCA score; we find that taking 1-1 features greatly increases this score from 0.03 to 0.35, as shown and explained in Figure 2.

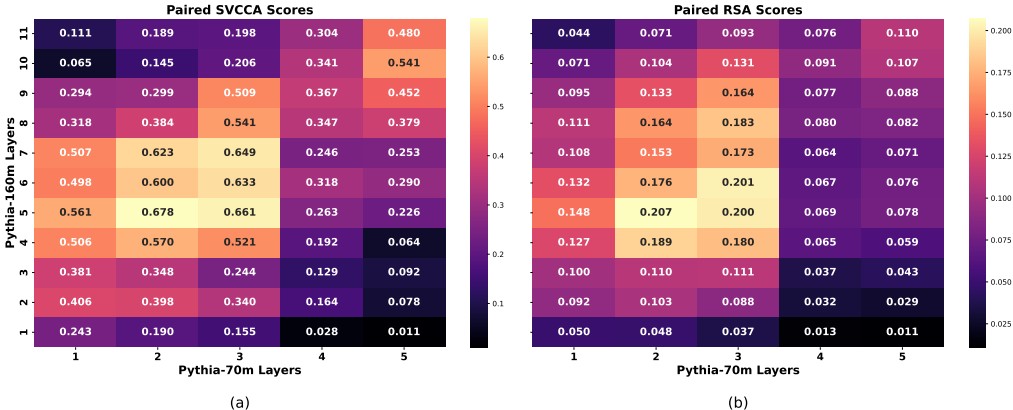

Figure 5: (a) SVCCA and (b) RSA **1-1** paired scores of SAEs for layers in Pythia-70m vs layers in Pythia-160m. Compared to Figure 4, some of the scores are slightly higher, and the SVCCA score at L2 vs L3 for 70m vs 160m is much higher. On the other hand, some scores are slightly lower, such as SVCCA at L5 vs L4 for 70m vs 160m. This is the same figure as Figure 2; it is shown here again for easier comparison to the Many-to-1 scores in Figure 4.

vs L4 for Pythia-70m vs Pythia-160m. Given that most layer pair scores are slightly higher, we use 1-1 feature mappings in our main text results.

Additionally, we notice that for semantic subspace experiments, 1-1 gave vastly better scores than many-to-1. We hypothesize that this is because since these feature subspaces are smaller, the noise from the many-to-1 mappings have a greater impact on the score.

We also looked into using more types of 1-1 feature matching, such as pairing many-to-1 features with their "next highest" or using efficient methods to first select the mapping pair with the highest correlation, taking those features off as candidates for future mappings, and continuing this for the next highest correlations. This also appeared to work well, though further investigation is needed. More analysis can also be done for mapping one feature from SAE A to many features in SAE B.

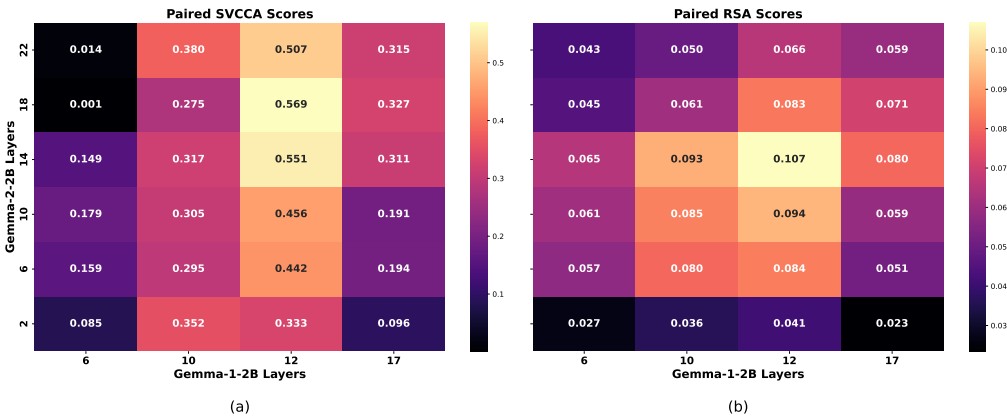

Figure 6: (a) SVCCA and (b) RSA **Many-to-1** paired scores of SAEs for layers in Gemma-1-2B vs layers in Gemma-2-2B. We obtain similar scores compared to the 1-1 paired scores in Figure 3.

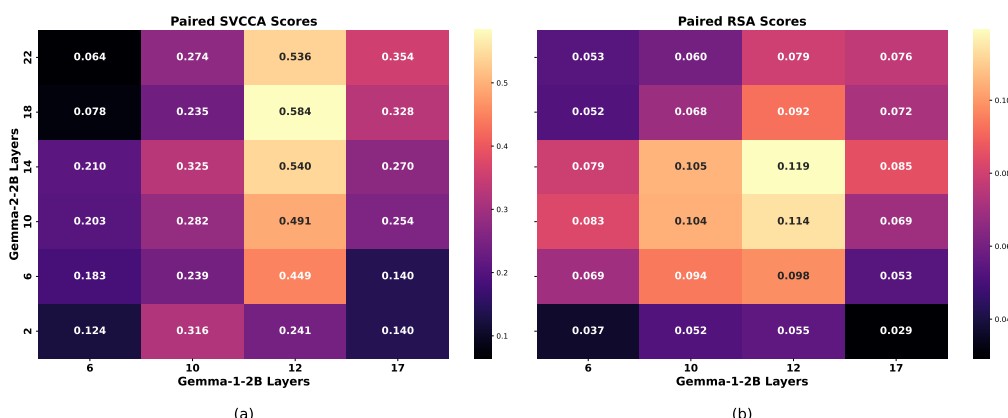

Figure 7: (a) SVCCA and (b) RSA **1-1** paired scores of SAEs for layers in Gemma-1-2B vs layers in Gemma-2-2B. Compared to Figure 6, some of the scores are slightly higher, whiel some are slightly lower. This is the same figure as Figure 3; it is shown here again for easier comparison to the Many-to-1 scores in Figure 4.

## B    NOISE OF NON-CONCEPT FEATURES

We define "non-concept features" as features which are not modeling specific concepts that can be mapped well across models. Their highest activations are on new lines, spaces, punctuation, and EOS tokens. As such, they may introduce noise when computing similarity scores, as their removal appears to greatly improve similarity scores. We hypothesize that one reason they introduce noise is that these could be tokens that aren't specific to concepts, but multiple concepts; thus, they are mixing concept spaces up when matching is done.

For instance, say feature $X$ from model $A$ fires highly on "!" in the sample "I am BOB!", while feature $Y$ from model $B$ fires highly on "!" in the sample, "that cat ate!"

A feature $X$ that activates on "!" in "BOB!" may be firing for names, while a feature $Y$ that activates on "!" in "ate!" may fire on food-related tokens. As such, features activating on non-concept features are not concept specific, and we could map a feature firing on names to a feature firing on food. We call these possibly "incoherent" mappings "non-concept mappings". By removing non-concept mappings, we show that SAEs reveal highly similar feature spaces.

We filter "non-concept features" that fire, for at least one of their top 5 dataset examples, on a "non-concept" keyword. The list of "non-concept" keywords we use is given below:

- \\n
- \n
- (empty string)
- (space)
- .
- ,
- !
- ?
- -
- <bos>
- <|endoftext|>

There may exist other non-concept features that we did not filter, such as colons, brackets and carats. We do not include arithmetic symbols, as those are domain specific to specific concepts. Additionally, we have yet to perform a detailed analysis on which non-concept features influence the similarity scores more than others; this may be done in future work.

## C  ILLUSTRATED STEPS OF COMPARISON METHOD

Figure 8 demonstrates steps 1 to 3 to carry out our similarity comparison experiments.

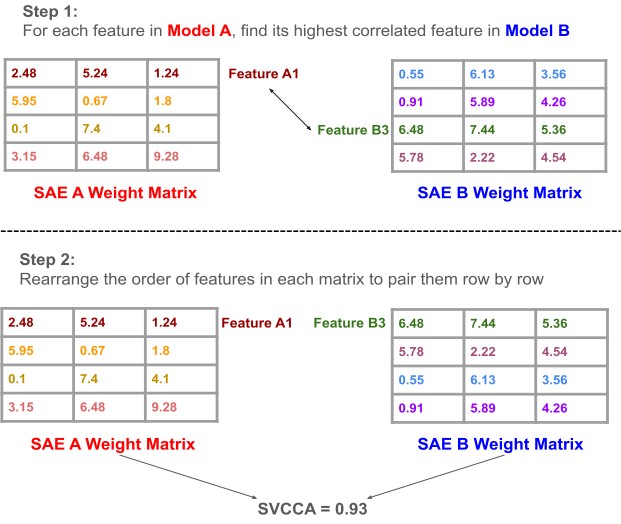

Figure 8: Steps to Main Results. We first find correlated pairs to solve neuron permutation, and then apply similarity metrics to solve latent space rotation issues.

## D  CONCEPT CATEGORIES LIST AND FURTHER ANALYSIS

The keywords used for each concept group for the semantic subspace experiments are given in Table 4. We generate these keywords using GPT-4 Bubeck et al. (2023) using the prompt "give a list of N single token keywords as a python list that are related to the concept of C", such that N is a number (we use N=50 or 100) and C is a concept like Emotions. We then manually filter these lists to avoid using keywords which have meanings outside of the concept group (eg. "even" does not largely mean

"divisible by two" because it is often used to mean "even if..."). We also aim to select keywords which are single-tokens with no spaces, and are case-insensitive. [3]

When searching for keyword matches from each feature's list of top 5 highest activating tokens, we use keyword matches that avoid features with dataset samples which use the keywords in compound words. For instance, when searching for features that activate on the token "king" from the People/Roles concept, we avoid using features that activate on "king" when "king" is a part of the word "seeking", as that is not related to the People/Roles concept.

We perform interpretability experiments to check which keywords of the features kept are activated on. We find that for many feature pairs, they activate on the same keyword and are monosemantic. Most keywords in each concept group are not kept. Additionally, for the layer pairs we checked on, after filtering there were only a few keywords that multiplied features fired on. This shows that the high similarity is not because the same keyword is over-represented in a space.

For instance, for Layer 3 in Pythia-70m vs Layer 5 in Pythia-160m, and for the concept category of Emotions, we find which keywords from the category appear in the top 5 activating tokens of each of the Pythia-70m features from the 24 feature mappings described in Table 1. We only count a feature if it appears once in the top 5 of a feature's top activating tokens; if three out of five of a feature's top activating tokens are "smile", then smile is only counted once. The results for Models A and B are given in Table 3. Not only are their counts similar, indicating similar features, but there is not a great over-representation of features. There are a total of 24 features, and a total of 25 keywords (as some features can activate on multiple keywords in their top 5). We note that even if a feature fires for a keyword does not mean that feature's purpose is only to "recognize that keyword", as dataset examples as just one indicator of what a feature's purpose is (which still can have multiple, hard-to-interpret purposes).

| Model A | | | Model B | |
|---|---|---|---|---|
| **Keyword** | **Count** | | **Keyword** | **Count** |
| free | 4 | | free | 5 |
| pain | 4 | | pain | 4 |
| smile | 3 | | love | 3 |
| love | 2 | | hate | 2 |
| hate | 2 | | calm | 2 |
| calm | 2 | | smile | 2 |
| sad | 2 | | kind | 1 |
| kind | 1 | | shy | 1 |
| shy | 1 | | doubt | 1 |
| doubt | 1 | | trust | 1 |
| trust | 1 | | peace | 1 |
| peace | 1 | | joy | 1 |
| joy | 1 | | sad | 1 |

Table 3: Count of Model A vs Model B features with keywords from the Emotion category in the semantic subspace found in Table 1.

However, as there are many feature pairs, we did not perform a thorough analysis of this yet, and thus did not include this analysis in this paper. We also check this in LLMs, and find that the majority of LLM neurons are polysemantic. We do not just check the top 5 dataset samples for LLM neurons, but the top 90, as the SAEs have an expansion factor that is around 16x or 32x bigger than the LLM dimensions they take in as input.

Calendar is a subset of Time, removing keywords like "after" and "soon", and keeping only "day" and "month" type of keywords pertaining to dates.

---

[3]However, not all the tokens in the Table 4 are single token.

Table 4: Keywords Organized by Category

| Category | Keywords |
|---|---|
| Time | day, night, week, month, year, hour, minute, second, now, soon, later, early, late, morning, evening, noon, midnight, dawn, dusk, past, present, future, before, after, yesterday, today, tomorrow, next, previous, instant, era, age, decade, century, millennium, moment, pause, wait, begin, start, end, finish, stop, continue, forever, constant, frequent, occasion, season, spring, summer, autumn, fall, winter, anniversary, deadline, schedule, calendar, clock, duration, interval, epoch, generation, period, cycle, timespan, shift, quarter, term, phase, lifetime, timeline, delay, prompt, timely, recurrent, daily, weekly, monthly, yearly, annual, biweekly, timeframe |
| Calendar | day, night, week, month, year, hour, minute, second, morning, evening, noon, midnight, dawn, dusk, yesterday, today, tomorrow, decade, century, millennium, season, spring, summer, autumn, fall, winter, calendar, clock, daily, weekly, monthly, yearly, annual, biweekly, timeframe |
| People/Roles | man, girl, boy, kid, dad, mom, son, sis, bro, chief, priest, king, queen, duke, lord, friend, clerk, coach, nurse, doc, maid, clown, guest, peer, punk, nerd, jock |
| Nature | tree, grass, stone, rock, cliff, hill, dirt, sand, mud, wind, storm, rain, cloud, sun, moon, leaf, branch, twig, root, bark, seed, tide, lake, pond, creek, sea, wood, field, shore, snow, ice, flame, fire, fog, dew, hail, sky, earth, glade, cave, peak, ridge, dust, air, mist, heat |
| Emotions | joy, glee, pride, grief, fear, hope, love, hate, pain, shame, bliss, rage, calm, shock, dread, guilt, peace, trust, scorn, doubt, hurt, wrath, laugh, cry, smile, frown, gasp, blush, sigh, grin, woe, spite, envy, glow, thrill, mirth, bored, cheer, charm, grace, shy, brave, proud, glad, mad, sad, tense, free, kind |
| MonthNames | January, February, March, April, May, June, July, August, September, October, November, December |
| Countries | USA, Canada, Brazil, Mexico, Germany, France, Italy, Spain, UK, Australia, China, Japan, India, Russia, Korea, Argentina, Egypt, Iran, Turkey |
| Biology | gene, DNA, RNA, virus, bacteria, fungus, brain, lung, blood, bone, skin, muscle, nerve, vein, organ, evolve, enzyme, protein, lipid, membrane, antibody, antigen, ligand, substrate, receptor, cell, chromosome, nucleus, cytoplasm |

Table 5: Number of Features in each Layer Pair Mapping after filtering Non-Concept Features for Pythia-70m (cols) vs Pythia-160m (rows) out of a total of 32768 SAE features in both models.

| Layer | 1 | 2 | 3 | 4 | 5 |
|---|---|---|---|---|---|
| 1 | 23600 | 21423 | 26578 | 19696 | 19549 |
| 2 | 16079 | 14201 | 17578 | 12831 | 12738 |
| 3 | 7482 | 7788 | 7841 | 7017 | 6625 |
| 4 | 12756 | 11195 | 13349 | 9019 | 9357 |
| 5 | 7987 | 6825 | 8170 | 5367 | 5624 |
| 6 | 10971 | 9578 | 10937 | 8099 | 8273 |
| 7 | 15074 | 12988 | 16326 | 12720 | 12841 |
| 8 | 14445 | 12580 | 15300 | 11779 | 11942 |
| 9 | 13320 | 11950 | 14338 | 11380 | 11436 |
| 10 | 9834 | 8573 | 9742 | 7858 | 8084 |
| 11 | 6936 | 6551 | 7128 | 5531 | 6037 |

Table 6: Number of **1-1** Features in each Layer Pair Mapping after filtering Non-Concept Features for Pythia-70m (cols) vs Pythia-160m (rows) out of a total of 32768 SAE features in both models.

| Layer | 1 | 2 | 3 | 4 | 5 |
|---|---|---|---|---|---|
| 1 | 7553 | 5049 | 5853 | 3244 | 3115 |
| 2 | 6987 | 4935 | 5663 | 3217 | 3066 |
| 3 | 4025 | 3152 | 3225 | 2178 | 1880 |
| 4 | 6649 | 5058 | 6015 | 3202 | 3182 |
| 5 | 5051 | 4057 | 4796 | 2539 | 2596 |
| 6 | 5726 | 4528 | 5230 | 3226 | 3031 |
| 7 | 7017 | 5659 | 6762 | 4032 | 3846 |
| 8 | 6667 | 5179 | 6321 | 3887 | 3808 |
| 9 | 6205 | 4820 | 5773 | 3675 | 3682 |
| 10 | 4993 | 3859 | 4602 | 3058 | 3356 |
| 11 | 3609 | 2837 | 3377 | 2339 | 2691 |

# E ADDITIONAL RESULTS

As shown in Figure 9 for Pythia and Figure 10 for Gemma, we also find that mean activation correlation does not always correlate with the global similarity metric scores. For instance, a subset of feature pairings with a moderately high mean activation correlation (eg. 0.6) may yield a low SVCCA score (eg. 0.03).

The number of features after filtering non-concept features and many-to-1 features are given in Tables 5 and 6. The number of features after filtering non-concept features and many-to-1 features are given in Tables 7 and 8.

Table 7: Number of Features in each Layer Pair Mapping after filtering Non-Concept Features for Gemma-1-2B (cols) vs Gemma-2-2B (rows) out of a total of 16384 SAE features in both models.

| Layer | 6 | 10 | 12 | 17 |
|---|---|---|---|---|
| 2 | 8926 | 8685 | 8647 | 4816 |
| 6 | 4252 | 4183 | 4131 | 2474 |
| 10 | 6458 | 6320 | 6277 | 3658 |
| 14 | 4213 | 4208 | 4214 | 2672 |
| 18 | 3672 | 3679 | 3771 | 2515 |
| 22 | 4130 | 4100 | 4168 | 3338 |

Table 8: Number of **1-1** Features in each Layer Pair Mapping after filtering Non-Concept Features for Gemma-1-2B (cols) vs Gemma-2-2B (rows) out of a total of 16384 SAE features in both models.

| Layer | 6 | 10 | 12 | 17 |
|---|---|---|---|---|
| 2 | 3427 | 3874 | 3829 | 1448 |
| 6 | 3056 | 3336 | 3261 | 1266 |
| 10 | 4110 | 4650 | 4641 | 1616 |
| 14 | 2967 | 3524 | 3553 | 1414 |
| 18 | 2636 | 3058 | 3239 | 1543 |
| 22 | 2646 | 3037 | 3228 | 1883 |

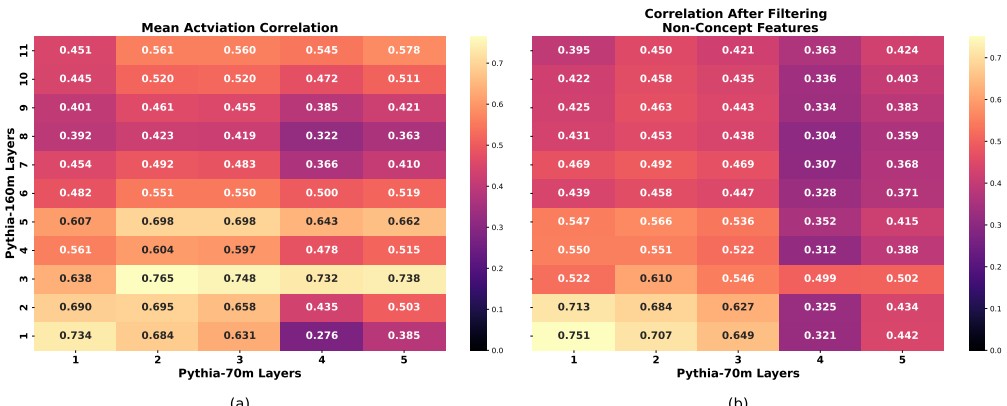

(a)                                                        (b)

Figure 9: Mean Activation Correlation before (a) and after (b) filtering non-concept features for Pythia-70m vs Pythia-160m. We note these patterns are different from those of the SVCCA and RSA scores in Figure 4, indicating that these three metrics each reveal different patterns not shown by other metrics.

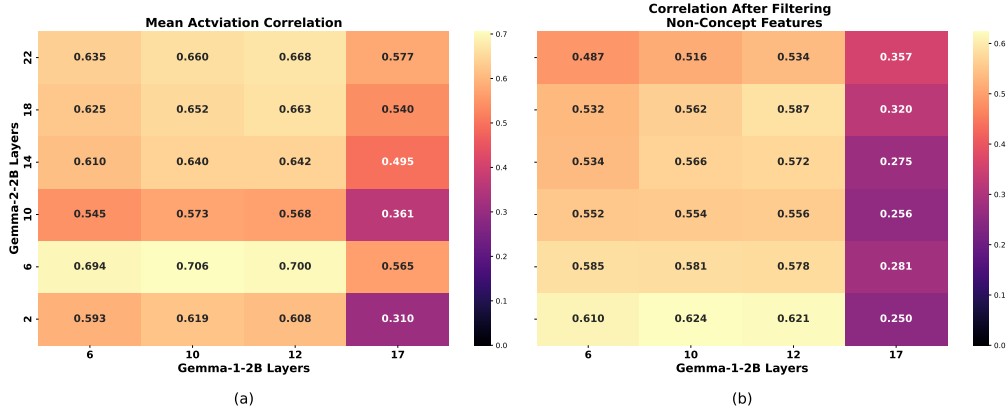

(a)                                                        (b)

Figure 10: Mean Activation Correlation before (a) and after (b) filtering non-concept features for Gemma-1-2B vs Gemma-2-2B. We note these patterns are different from those of the SVCCA and RSA scores in Figure 6, indicating that these three metrics each reveal different patterns not shown by other metrics.

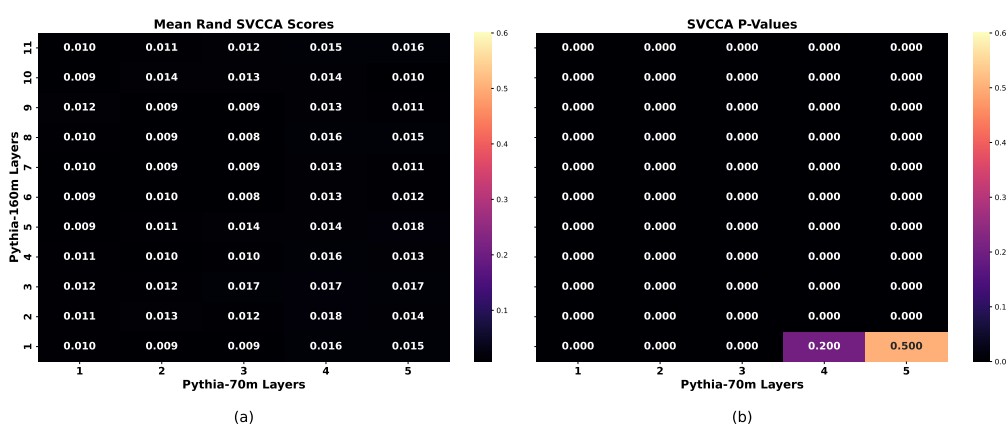

Figure 11: Mean Randomly Paired SVCCA scores and P-values of SAEs for layers in Pythia-70m vs Pythia-160m. Compared to Paired Scores in Figure 4, these are all very low.

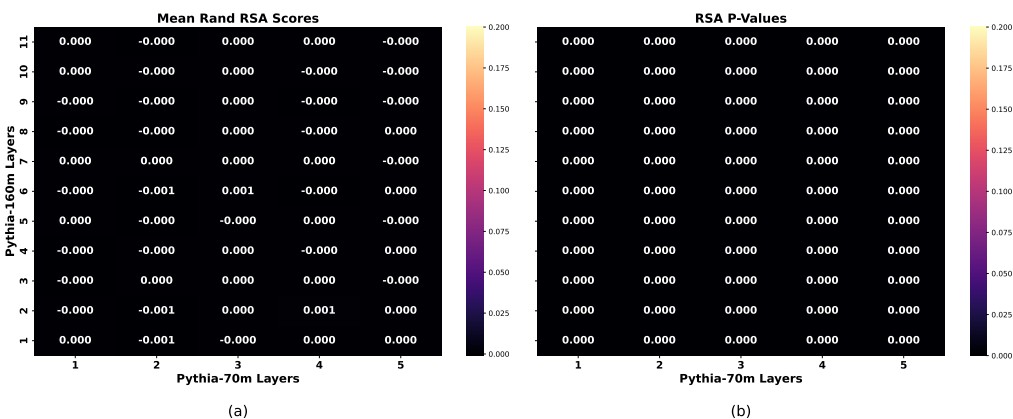

Figure 12: Mean Randomly Paired RSA scores and P-values of SAEs for layers in Pythia-70m vs Pythia-160m. Compared to Paired Scores in Figure 4, these are all very low.

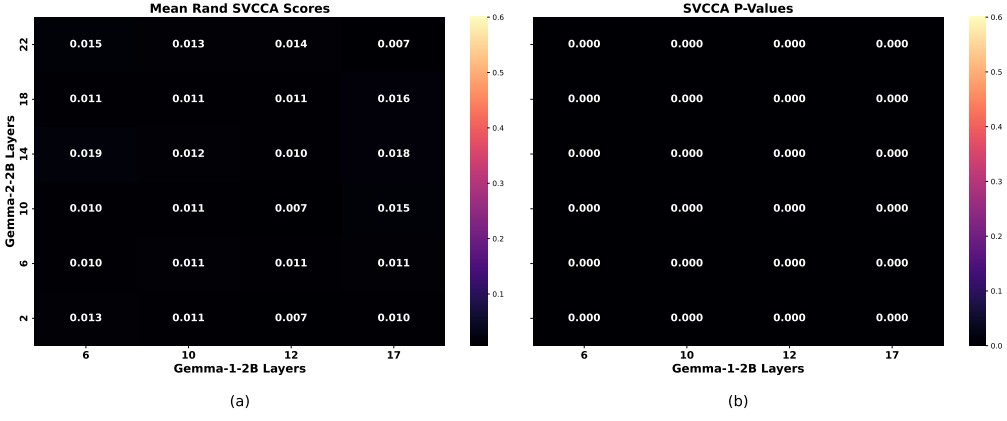

Figure 13: Mean Randomly Paired SVCCA scores and P-values of SAEs for layers in Gemma-1-2B vs layers in Gemma-2-2B. Compared to Paired Scores in Figure 6, these are all very low.

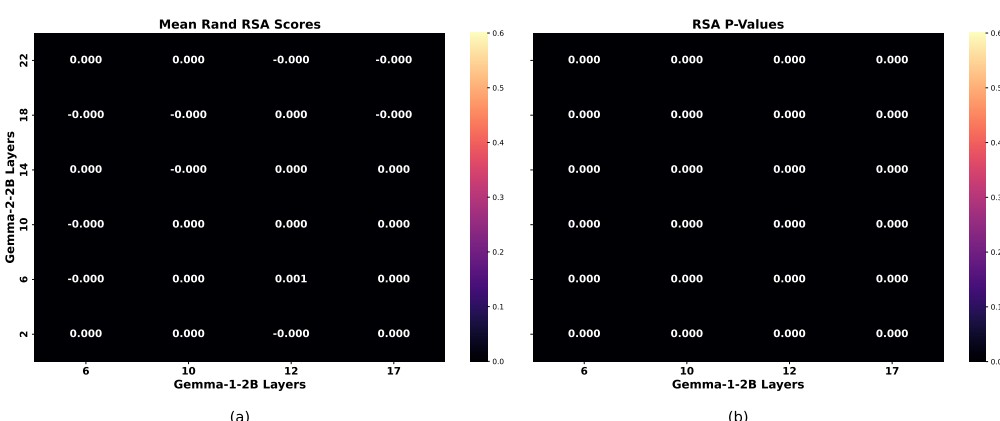

(a)                    (b)

Figure 14: Mean Randomly Paired RSA scores and P-values of SAEs for layers in Gemma-1-2B vs layers in Gemma-2-2B. Compared to Paired Scores in Figure 6, these are all very low.

Table 9: RSA scores and random mean results for 1-1 semantic subspaces of L3 of Pythia-70m vs L5 of Pythia-160m. P-values are taken for 1000 samples in the null distribution.

| Concept Subspace | Number of 1-1 Features | Paired Mean | Random Shuffling Mean | p-value |
|---|---|---|---|---|
| Time | 228 | 0.10 | 0.00 | 0.00 |
| Calendar | 126 | 0.09 | 0.00 | 0.00 |
| Nature | 46 | 0.22 | 0.00 | 0.00 |
| MonthNames | 32 | 0.76 | 0.00 | 0.00 |
| Countries | 32 | 0.10 | 0.00 | 0.03 |
| People/Roles | 31 | 0.18 | 0.00 | 0.01 |
| Emotions | 24 | 0.46 | 0.00 | 0.00 |

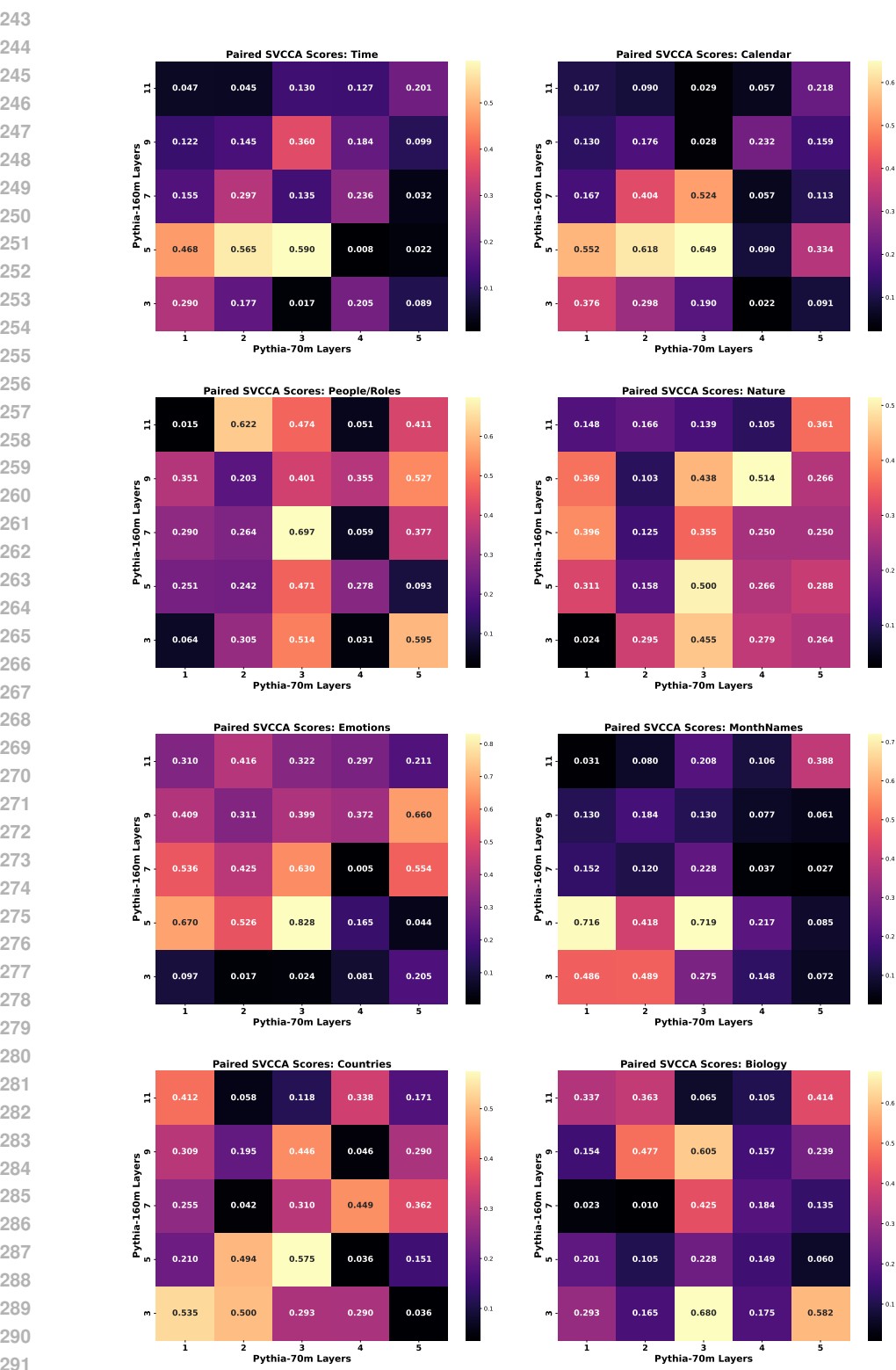

Figure 15: 1-1 Paired SVCCA scores of SAEs for layers in Pythia-70m vs layers in Pythia-160m for Concept Categories. Middle layers appear to be the most similar with one another.

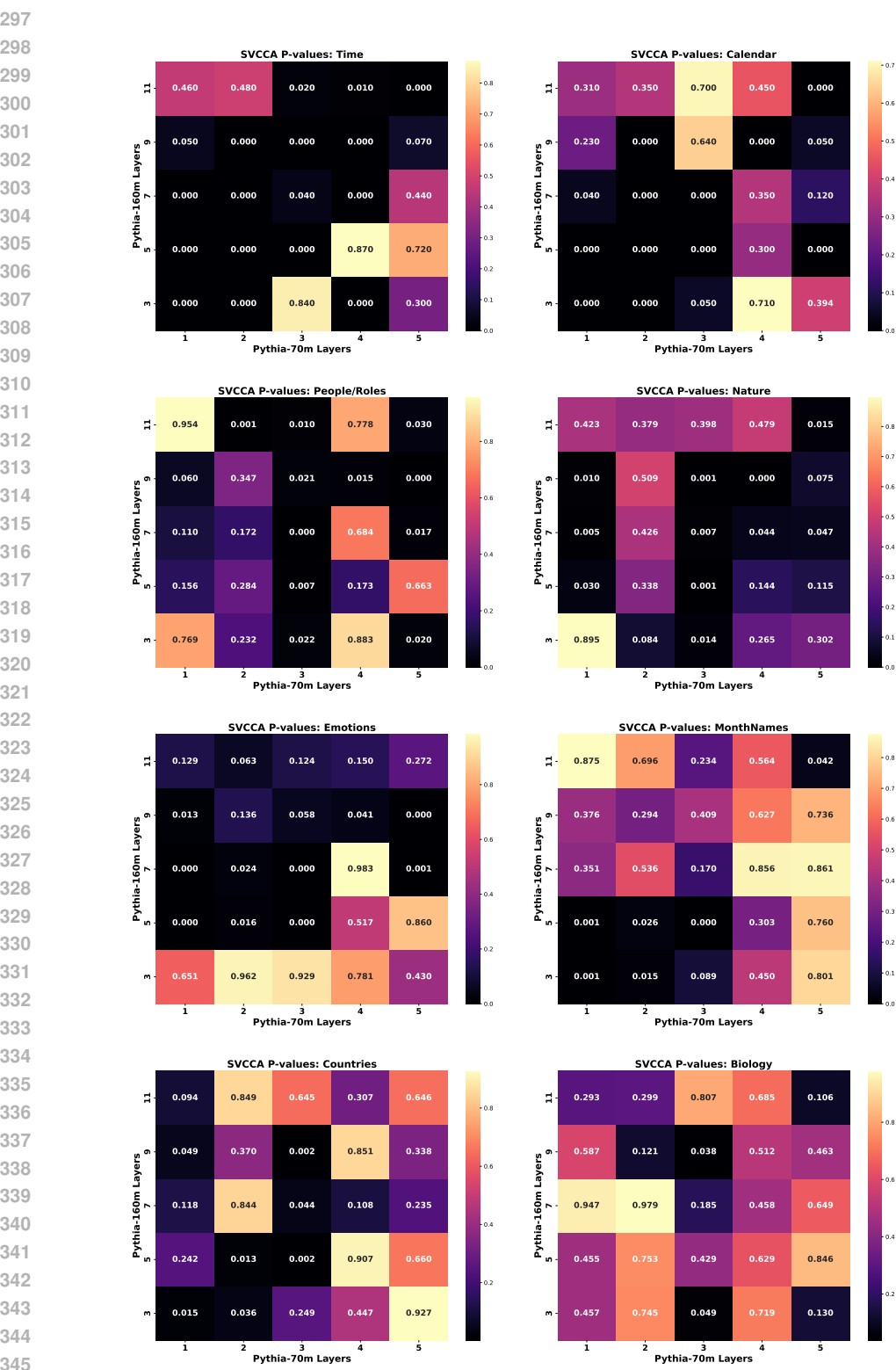

Figure 16: 1-1 SVCCA p-values of SAEs for layers in Pythia-70m vs layers in Pythia-160m for Concept Categories. A lower p-value indicates more statistically meaningful similarity.

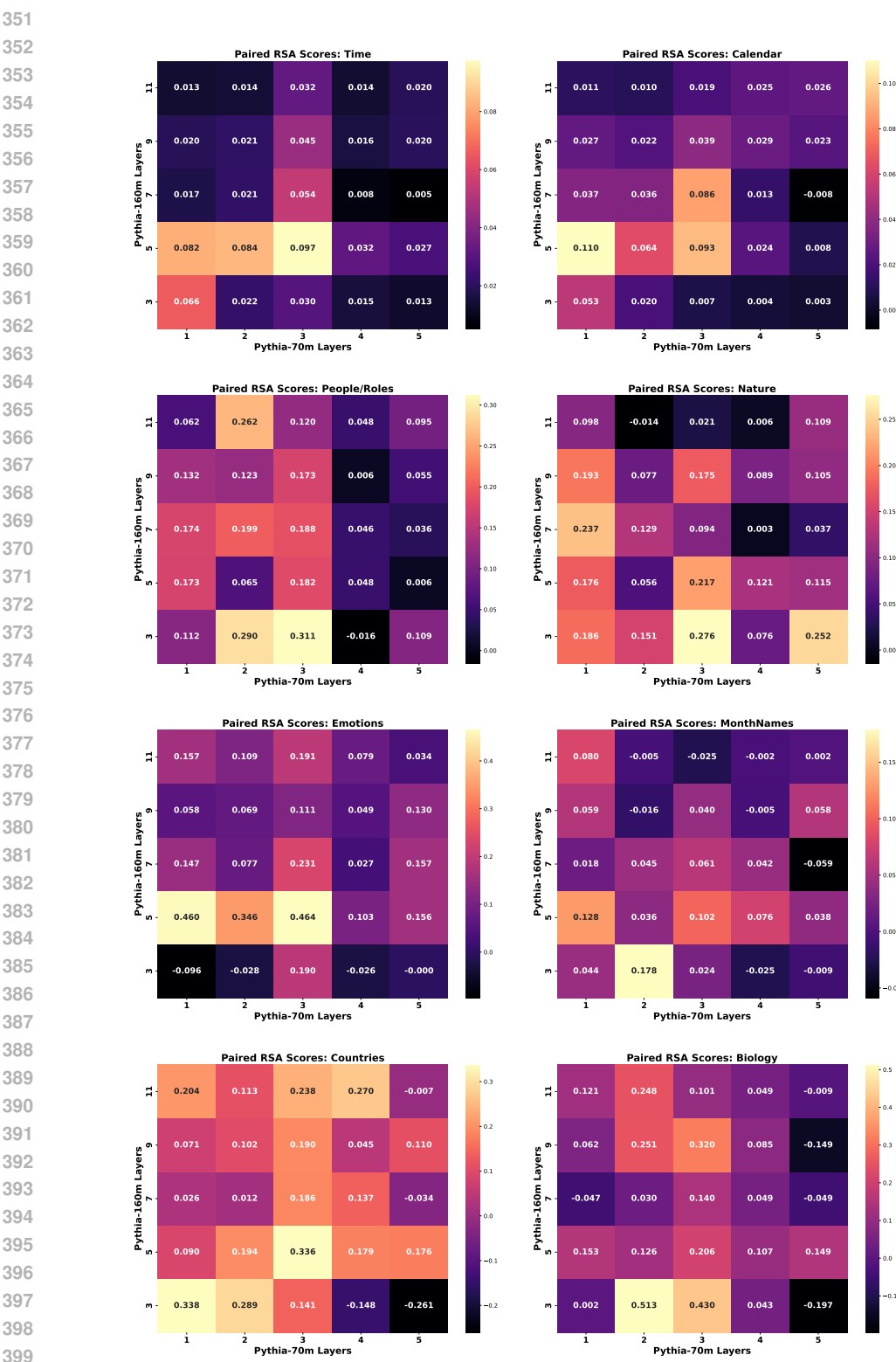

Figure 17: 1-1 Paired RSA scores of SAEs for layers in Pythia-70m vs layers in Pythia-160m for Concept Categories. Middle layers appear to be the most similar with one another.

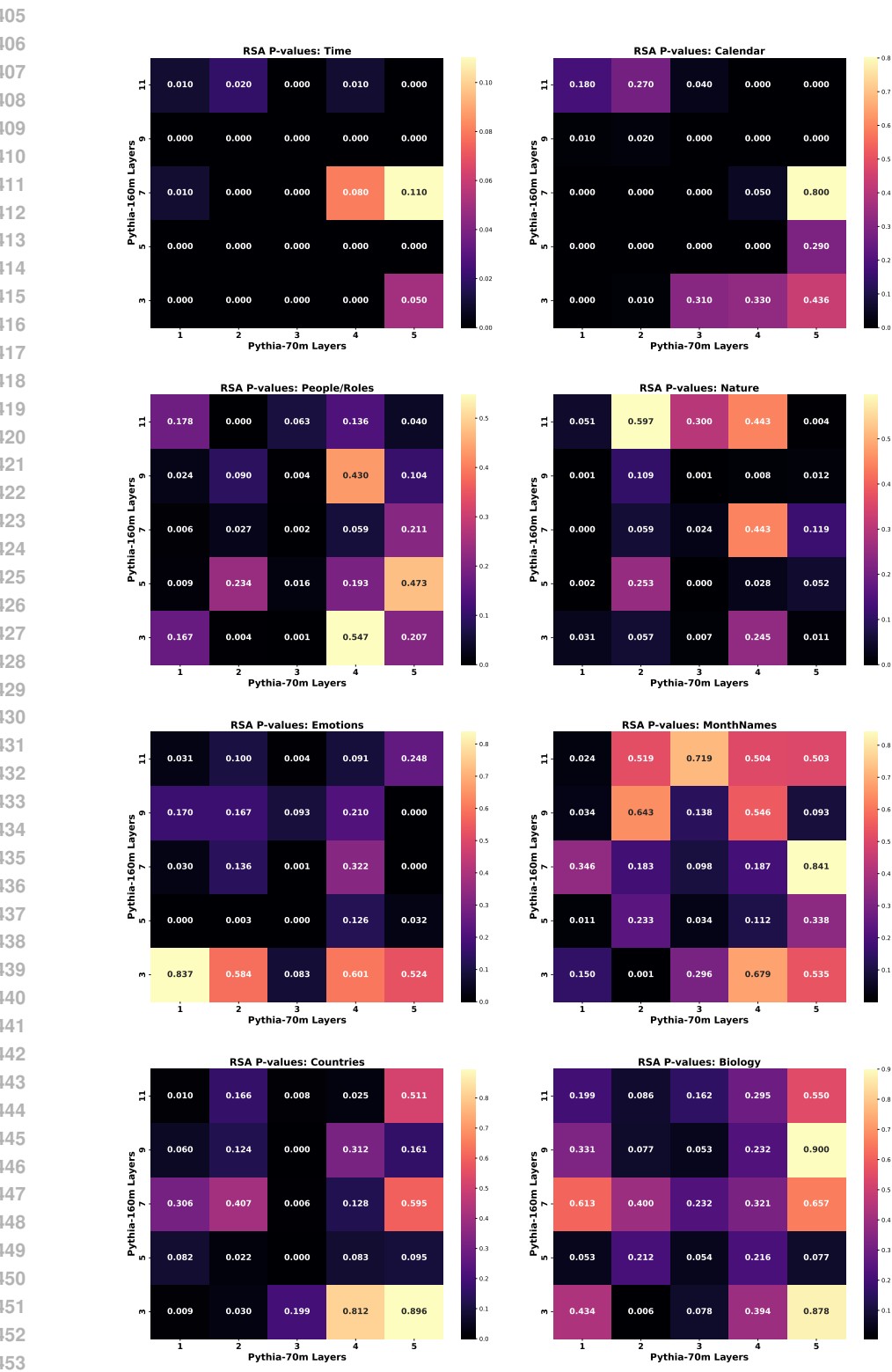

Figure 18: 1-1 RSA p-values of SAEs for layers in Pythia-70m vs layers in Pythia-160m for Concept Categories. A lower p-value indicates more statistically meaningful similarity.

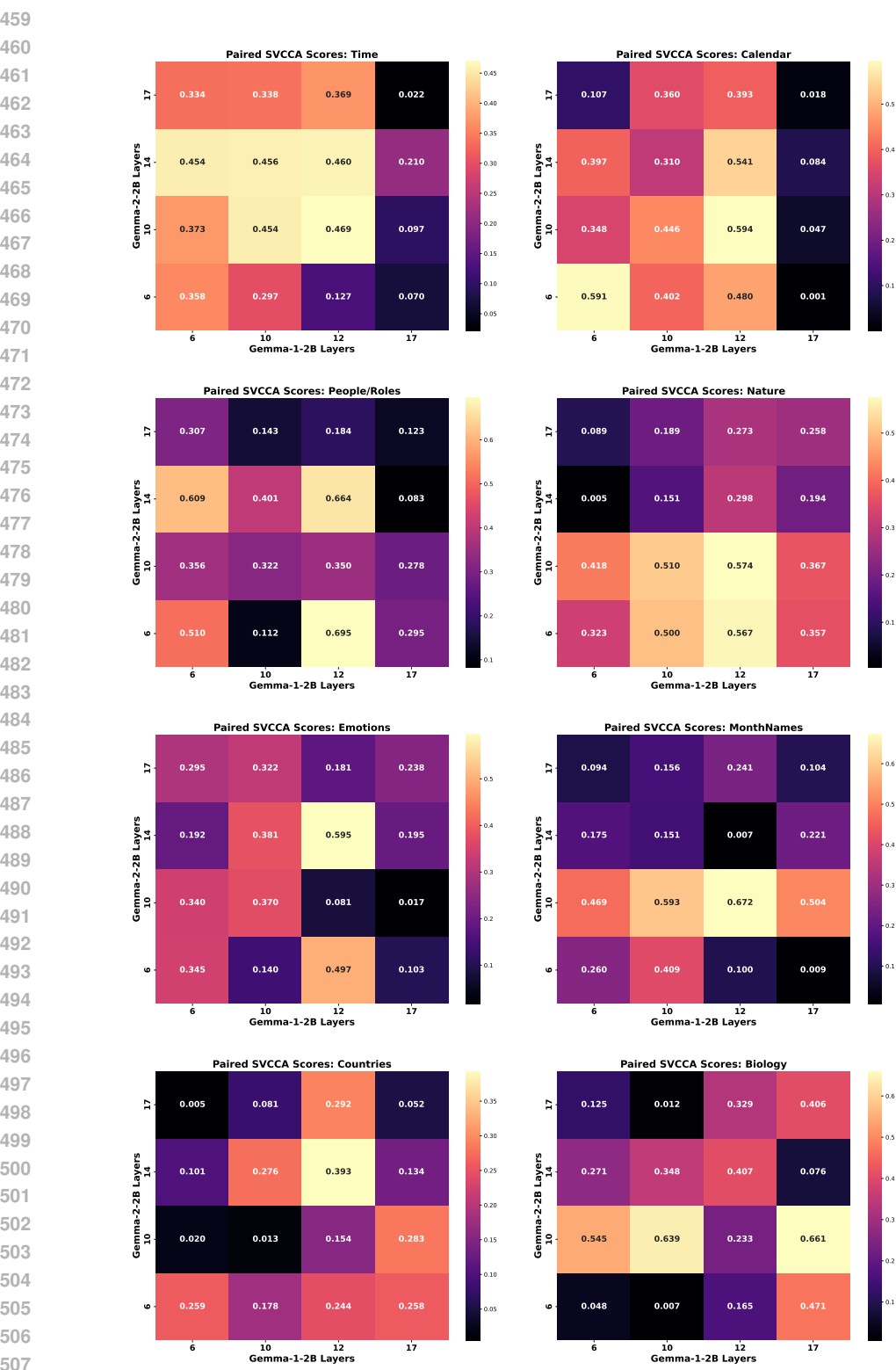

Figure 19: 1-1 Paired SVCCA scores of SAEs for layers in Gemma-1-2b vs layers in Gemma-2-2b for Concept Categories. Middle layers appear to be the most similar with one another.

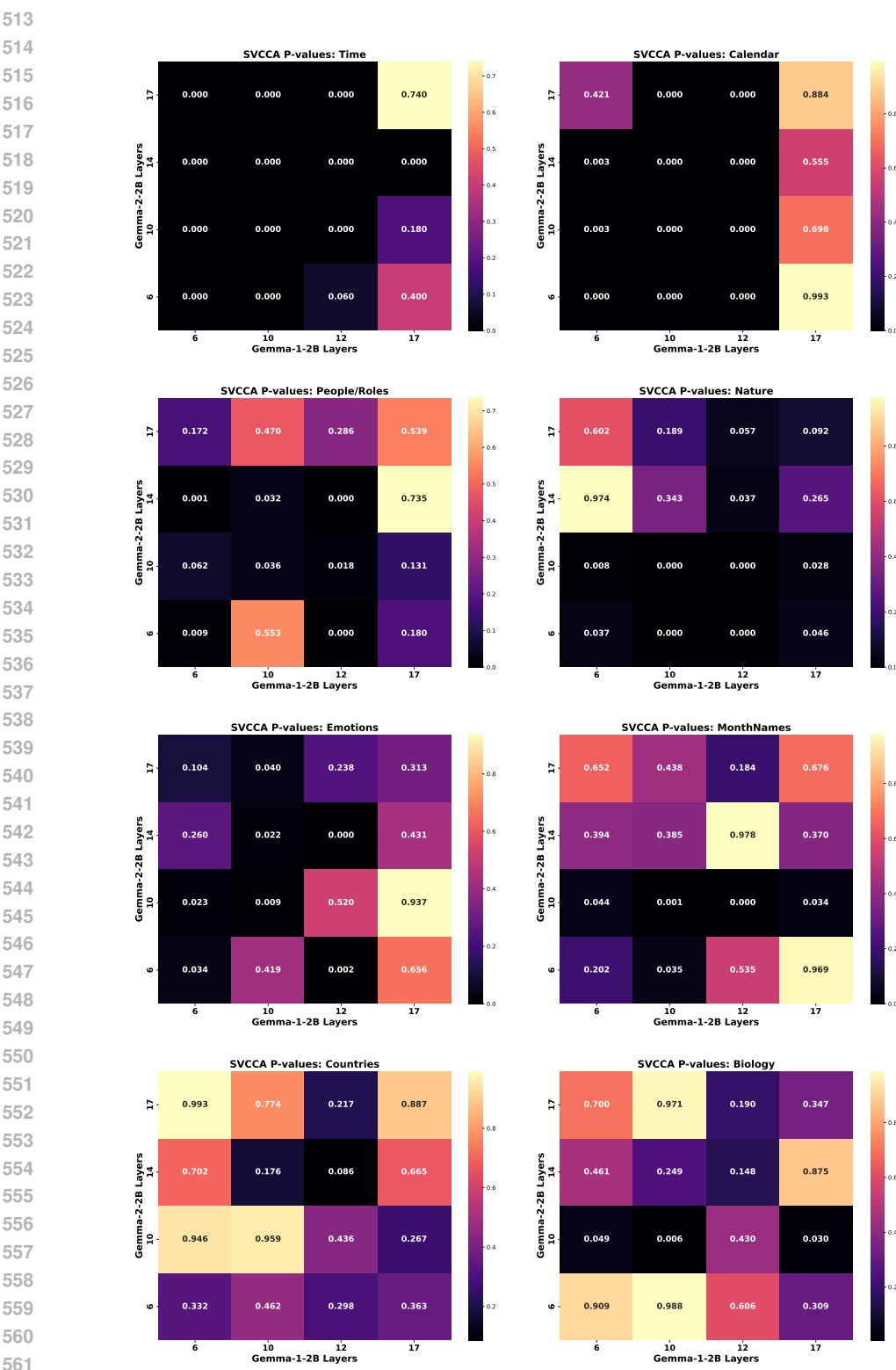

Figure 20: 1-1 SVCCA p-values of SAEs for layers in Gemma-1-2b vs layers in Gemma-2-2b for Concept Categories. A lower p-value indicates more statistically meaningful similarity.

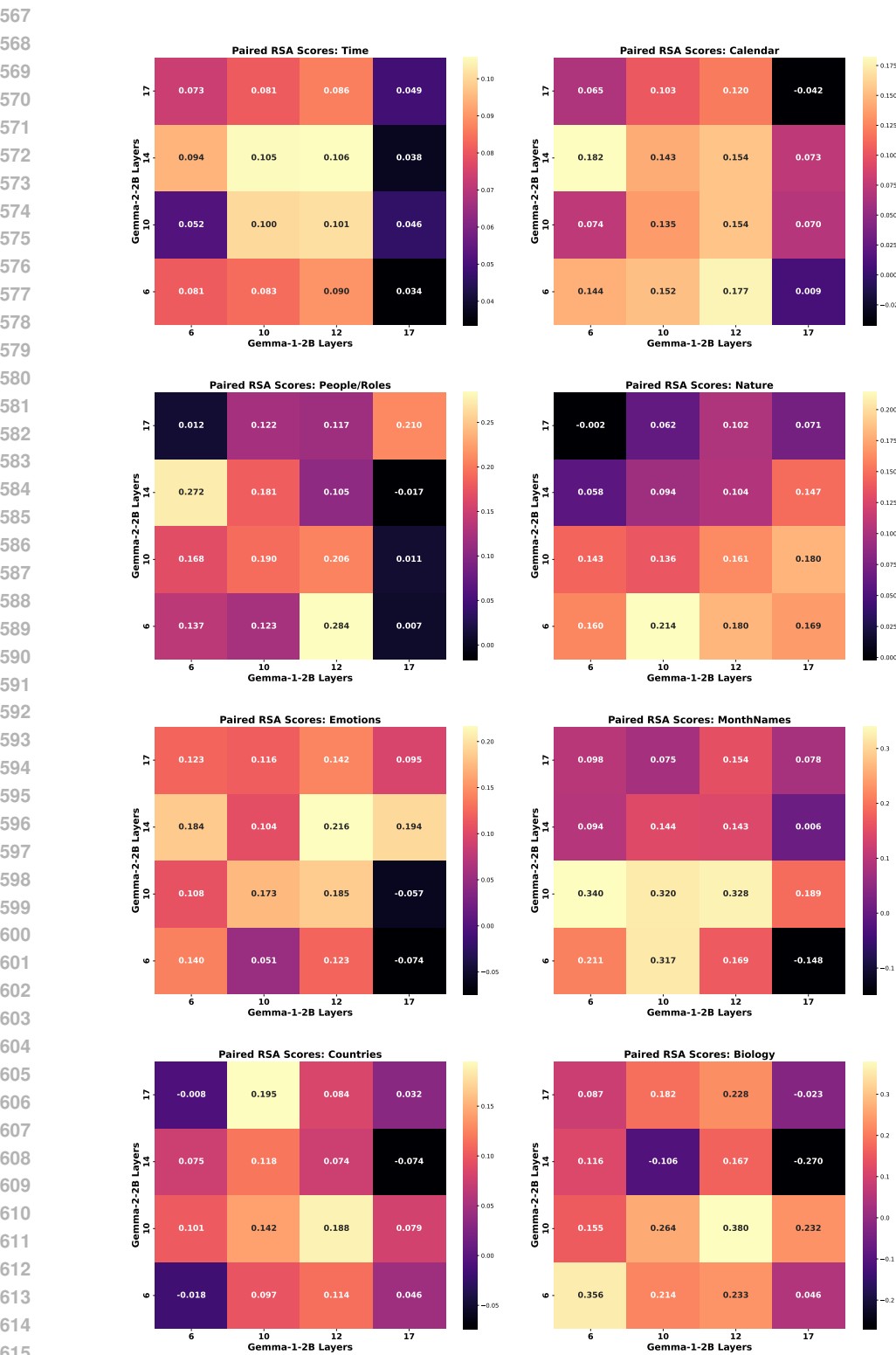

Figure 21: 1-1 Paired RSA scores of SAEs for layers in Gemma-1-2b vs layers in Gemma-2-2b for Concept Categories. Middle layers appear to be the most similar with one another.

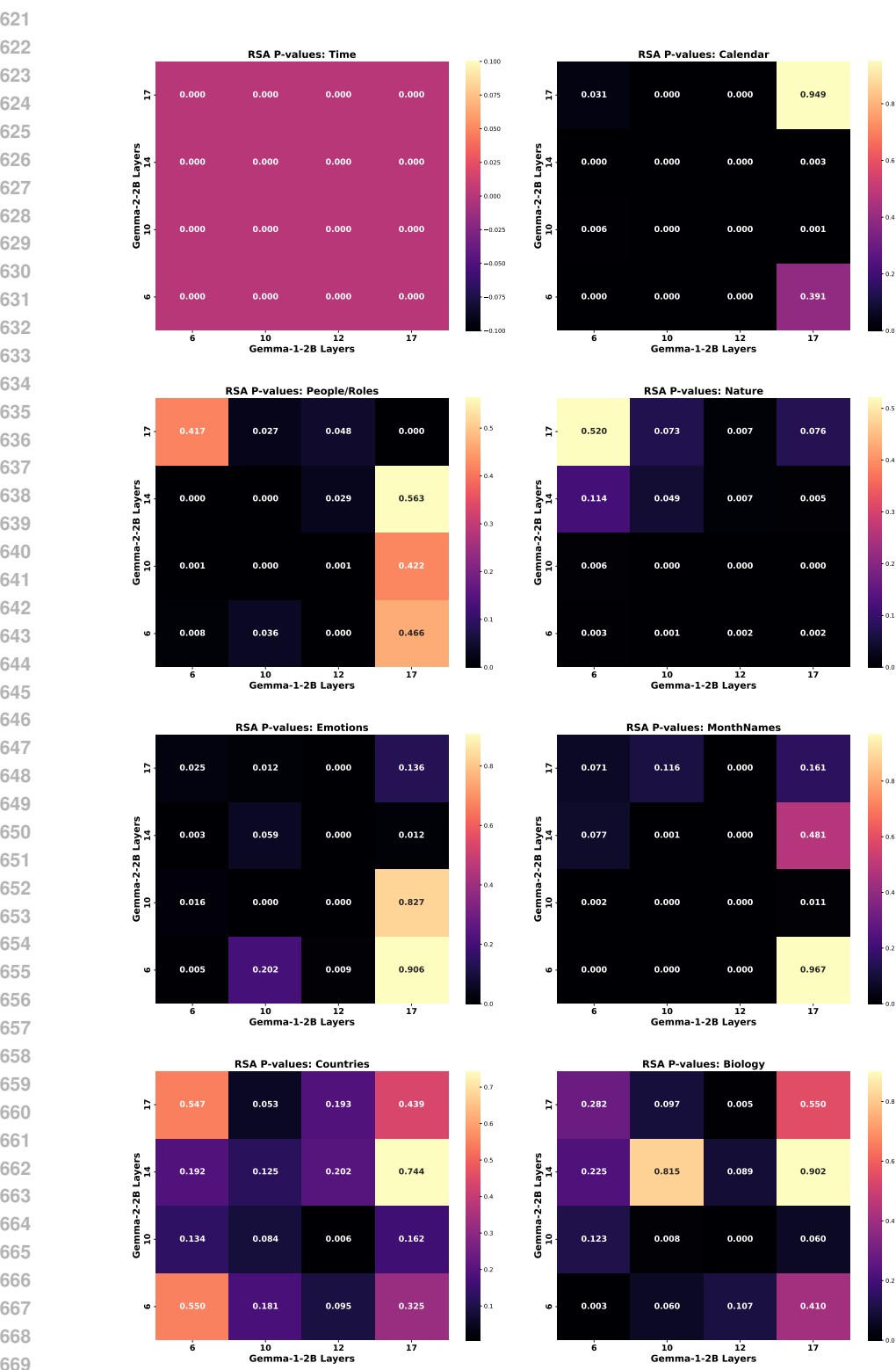

Figure 22: 1-1 RSA p-values of SAEs for layers in Gemma-1-2b vs layers in Gemma-2-2b for Concept Categories. A lower p-value indicates more statistically meaningful similarity.

