# OpenReview forum: "Sparse Autoencoders Reveal Universal Feature Spaces Across Large Language Models"
_ICLR.cc/2025/Conference — Submitted to ICLR 2025_

### Official Review · Reviewer_gv83 · 2024-10-31

**Soundness:** 2
**Presentation:** 3
**Contribution:** 3
**Rating:** 6
**Confidence:** 3

**Summary:**

This paper aims to explore whether latent features obtained with sparse autoencoders (SAEs) are shared across different sizes within LLM families (Pythia and Gemma). The authors match individual features and broader feature spaces using rotation invariant methods.

The authors match the most correlated feature pairs and run statistical tests to compare them with random feature pairs to obtain a p-value.

My **recommendation**: Accept: 6/10 (borderline paper).

### Why not lower:
- The correlations authors find are notable.
- This paper’s methodology could be useful in answering many related research questions.

### Why not higher
- Some experiments would benefit from a stronger baseline.
- While interesting, the framing of the research question within the literature is somewhat limited.

**Strengths:**

## Statistical significance

The correlations found seem very statistically **significant**, and some spaces have somewhat high correlations (around 0.6). This suggests that there are indeed shared latent feature spaces across model sizes. This is a **novel** insight into scaling dynamics. The authors further find interesting correlations between individual features of 0.2 on average, which indicates some individual features are replicated across models.

## Strong correlation between known spaces

The authors include what I understand to be an analysis of the correlation between known feature spaces (ex. time nature emotions) in Table 1. This indicates that the correlations are of high **quality**, as we know the ground truth.

## Lagged correlations

As model size scales and more layers are added, the highest correlations in gemma models are between spaces early in the smaller model and later in the bigger model. This is an interesting insight, as it suggests that the increased number of parameters could be spent in the earlier layers computing (possibly) low level features. This could be an interesting future direction.

## Feature transfer in AI alignment.

The implication that known feature spaces are preserved across model scale is **significant** for broader AI alignment goals, such as the possibility to transfer alignment techniques from a model of generation N to the next generation N+1.

**Weaknesses:**

## Weak baseline

Authors take random feature pairs as a baseline. This is not ideal as a baseline. Comparing the most correlated pair of activations with random pairs is expected to yield low p-values. If you have 100 random pairs of activations (here there are more), and you take the most correlated pair it will yield a p-value of 0.01. Here there are many more, and some are bound to be (spuriously?) correlated.

Authors may take the most correlated pairs of features in models that have nothing in common as a baseline. If the high correlations still show up, then they are spurious and caused by the sorting itself. An example of models with nothing (or little) in common might be LLMs doing very specialized tasks in different languages. Alternatively, the authors could inspect the relevant pairs to see if the correlations are genuine, which I see is somewhat done by taking the mean correlations of known features in Table 1. This somewhat eases my concerns with the methodology.

## Lack of comparison between families

While this is acknowledged, and comparison between different model scales is very interesting, only comparing models within the same family somewhat limits the scope of the findings. It would be interesting to inspect how similar families are (for ex. a big Pythia vs Gemma 2B).

## Few model families

As far as I have read, the authors compare some layers for 2 models from each of the Pythia/Gemma model families. While it is acknowledged that this is due to there only being 2 model families for which open-source SAEs are available, they are still a somewhat limited sample of SAEs to experiment on. While there are no other model families for which different sizes have SAEs, there are other LLMs for which there are SAEs, and it would be interesting to experiment on them.

**Questions:**

Have you tried the Llama SAE?

Have you tried more layers?

Have you tried comparing across families, for example, a large Pythia with Gemma 2B?

Have you tried comparing unrelated models as a baseline?

---

> ### Author Response · Authors · 2024-11-21
> **Response to Reviewer gv83**
>
> We thank the reviewer for their comments and will address their points in our response.
>
> > While interesting, the framing of the research question within the literature is somewhat limited.
>
> There is limited literature on the universality of SAE features so far, as it is an early area of research and only Bricken et al. (2023) [1] have investigated this. We seek to take the first steps to explore this new area.
>
> [1] https://transformer-circuits.pub/2023/monosemantic-features/index.html
>
> > Authors take random feature pairs as a baseline. This is not ideal as a baseline.
>
> This baseline was used as an effective baseline by Kriegeskorte et al. (2008), which developed the RSA technique. It is of note that taking the most correlated feature pairings, even those with high mean activation correlation, does not always yield the highest SVCCA or RSA scores. For instance, Figure 3a shows that Layer 6 vs Layer 22 for Gemma-1-2B vs Gemma-2-2B yields a low SVCCA score of 0.06, despite a high mean activation correlation of 0.635 as shown in Figure 10a.
>
> > Authors may take the most correlated pairs of features in models that have nothing in common as a baseline… An example of models with nothing (or little) in common might be LLMs doing very specialized tasks in different languages.
>
> This is an interesting approach, though there are no pretrained SAEs on very specialized models available so we would have to train these for an updated version of the paper later.
>
> > It would be interesting to inspect how similar families are (for ex. a big Pythia vs Gemma 2B).
>
> We are currently working on these experiments to include in an updated version of the paper. Cross-model family experiments require a new approach as our current approach for SAE feature pairings rely on token-wise activation correlations, necessitating all models to share the same tokenizer.
>
> > While there are no other model families for which different sizes have SAEs, there are other LLMs for which there are SAEs, and it would be interesting to experiment on them.
>
> As mentioned before, since our experiments utilize activation correlation by token, which requires each model to use the same tokenizer, we only select model pairs if there exist SAEs trained on models that use the same tokenizer.

---

> > ### Comment · Reviewer_gv83 · 2024-11-25
> >
> > Dear authors,
> >
> > Thanks for your response.
> > I understand that no other full model SAEs are available, but it would still be interesting to see just one or two layers from specialised SAEs as a baseline (this should not be too expensive).
> >
> > I would be very happy to see comparisons between families in a revision of this paper.
> >
> > If these concerns were addressed I would increase my score to 8.

---

> > > ### Author Response · Authors · 2024-11-25
> > >
> > > We thank the reviewer again for their review. We will try to run experiments on "one or two layers from specialised SAEs as a baseline", though due to remaining time constraints this may not be feasible. Namely, we are not aware of two, small open source models that have "nothing in common as a baseline", such as those that do "very specialized tasks in different languages". We are interested in finding these and would appreciate suggestions that the reviewer is aware of. If not, we will look to train these LLMs, and then train SAEs on them, though this will most likely take longer than the remaining available review time.
> > >
> > > In terms of smaller SAEs on a "specialized" dataset, we have also found high similarities between Tinystories-1-Layer (21M parameters) and Tinystories-2-Layer (33M parameters), though these models are very similar.

---

> > > > ### Comment · Reviewer_gv83 · 2024-11-25
> > > >
> > > > Dear authors.
> > > >
> > > > I meant different specialised datasets not the two models working on the same specialised dataset, since this should be a lower bound for correlations. A example is https://huggingface.co/m-a-p/CT-LLM-Base a 2B parameters LM in chinese with https://huggingface.co/budecosystem/code-millenials-1b a 1B coding model.
> > > >
> > > > I understand there is not enough time to include comparisons between families, or on specialised models as a baseline, and I think this reduces the scope of the contributions, but I believe it should still be accepted.

---

> > > > > ### Author Response · Authors · 2024-11-25
> > > > >
> > > > > We thank the reviewer for their suggested models. We will look into seeing how feasible it is to train and compare SAEs on these two models within the remaining review time.

---

> > > > > > ### Author Response · Authors · 2024-11-30
> > > > > >
> > > > > > To update on this topic, as described in the discussion about cross-family model comparisons, we have been working on approaches comparing different tokenizers, but so far find that this is a non-trivial problem that may not be solved without significant chances to our paper’s main approach; this may also require vast rewrites to our paper. Though we were interested in using the Chinese-centric CT-LLM-Base, we find that it uses a different tokenizer than other models. Thus, we have been searching for other specialized models that still use the same tokenizer; however, it appears many existing specialized models each use different tokenizers. As of now, the best approach to comparing individual features is based on Bricken et al. (2023) [1], from which our paper builds upon in a novel way.
> > > > > >
> > > > > > We are exploring and brainstorming other alternative baselines, and are open to any suggestions about other alternative baselines that the reviewer may have. We also inquire as to what requirements the reviewer has in mind for this baseline, based on what its main purpose is. For instance, is this baseline’s main purpose to ensure unrelated models have low similarity, thus showing that the measures should produce low or high similarity in expected cases? We note that far away layers in models do have low similarity (as shown in the top right corner of Figure 6a), while middle layers have high similarity to one another. Also of note when discussing what is a suitable additional baseline is that it may be difficult to train high-quality SAEs from scratch in the remaining review period.
> > > > > >
> > > > > > [1] https://transformer-circuits.pub/2023/monosemantic-features/index.html

---

### Official Review · Reviewer_es5D · 2024-11-02

**Soundness:** 3
**Presentation:** 3
**Contribution:** 4
**Rating:** 8
**Confidence:** 4

**Summary:**

This paper investigates feature universality in large language models by analysing similarities between sparse autoencoder (SAE) representations across different models. The authors introduce a methodology for comparing SAE feature spaces using correlation-based feature matching followed by rotation-invariant similarity metrics (SVCCA and RSA). The work presents evidence for universal features across model scales within architecture families and finds particular similarity in semantic subspaces and middle layers.

**Strengths:**

1. Clear methodology for comparing SAE spaces, with careful handling of both permutation and rotation invariance issues
2. Systematic approach to noise reduction through filtering non-concept features and many-to-1 mappings
3. Results show consistent patterns across model scales, particularly in middle layers
4. Findings about middle layer similarity align with existing work on model steering
5. Results on semantic subspace similarity provide interesting insights into how models represent concepts

Overall, I think it's a really important piece of work that was sorely needed. Universal feature spaces provides some good evidence for a weak form of the linear representation hypothesis, which may give us encouragement that SAEs are the right level of abstraction to think about atomic units of model computation.

**Weaknesses:**

1. Limited comparison to alternative methods. The paper does not consider using Maximum Correlation Coefficient (MCC) with Hungarian matching on either decoder weights or latent activations. This would provide a simpler, provably optimal matching baseline against which to compare their approach. (I know MCC is optimal up to sign flips and rotations, but I'd still like to see how this looks as a very simple approach.)

2. Experimental limitations:
   - Comparisons limited to within-family pairs (Pythia and Gemma)
   - Limited cross-architecture analysis to support broader universality claims (however, I understand they only compared MLPs because they couldn't properly compare residual streams across architectures due to different ways of combining MLP and attention outputs)
   - Small dataset size (100 samples) for statistical testing
   - Limited ablation studies on filtering choices

3. Methodological questions:
   - Selection of non-concept tokens appears somewhat arbitrary
   - Choice of semantic categories lacks rigorous validation
   - Top-5 activation analysis may be insufficient for feature characterisation - typically, people sample a spectrum of activations and analyse these e.g. see Templeton et al. (2023). There is inherent bias in looking at only the top activating samples
   - Statistical significance testing could be more rigorous (multiple hypothesis testing not addressed)

4. Missing details:
   - SAE training parameters
   - Computational requirements and scalability
   - Complete implementation details for similarity metrics
   - Thresholds for filtering decisions
   - You're missing a couple of particularly relevant citations. Others have explored similar feature alignment studies, particularly across different SAEs. For instance, see https://arxiv.org/abs/2408.00657#:~:text=Sparse%20autoencoders%20(SAEs)%20have%20shown,effectiveness%20in%20disentangling%20semantic%20concepts.


MINOR COMMENTS:

1. Abstract claims "rigorous" analysis - let readers judge this
2. "Comprehensive experiments" overstates the scope of testing
3. Some figures could be more clearly labeled
4. Statistical significance discussion needs more detail

**Questions:**

1. Have you compared your correlation-based matching to MCC with Hungarian matching (on either decoder weights or latent activations)? This would provide a principled baseline.

2. How sensitive are the results to:
   - Choice of non-concept tokens for filtering?
   - Number of top activations used for feature analysis?
   - Feature neuron count in the SAEs?

3. Would the findings generalise across different architecture families? Cross-architecture comparisons would strengthen universality claims. I'd even be interested to see the results repeated with something like Llama, which I understand has a bit of a different setup to Gemma and Pythia, even in things like positional embeddings. If you could replicate these findings, it would be good

4. Can you include a bit more of a general discussion on the linear representation hypothesis? I know it can become difficult to not be too philosophical, but this is essentially what you're testing. Giving the reader some grounding on why we think SAE features might be able to learn the atomic units of model computation really shapes the contours of your work.

---

> ### Author Response · Authors · 2024-11-21
> **Response to Reviewer es5D**
>
> We thank the reviewer for their comments and score, and will address their points in our response.
>
> > Limited comparison to alternative methods.
> > Comparisons limited to within-family pairs (Pythia and Gemma)
> > Small dataset size (100 samples) for statistical testing
>
> We are currently working on these experiments. Additionally, we note that though there were only 100 samples, each sample had 100 tokens, so there were a total of 100k activations.
>
> > Selection of non-concept tokens appears somewhat arbitrary
> This selection was chosen based on interpretability of frequently occurring features. We noticed that there were a large number of features that activated for non-concept tokens such as <endoftext> and \n (newlines), and that when we removed these features, the model performance greatly improved.
>
> > Choice of semantic categories lacks rigorous validation
>
> Though this is an important point, it is not likely a feasible task for this paper, as there is no ground truth labeling of semantic categories to allow for rigorous validation.
>
> > Missing details and minor comments
> We will revise the paper to incorporate these suggestions.
> > Can you include a bit more of a general discussion on the linear representation hypothesis?
>
> We include this in an updated version of the paper in the Appendix, and will cite relevant work in our discussion.

---

> > ### Author Response · Authors · 2024-11-25
> >
> > We thank the reviewer again for their review. We have posted new responses that have aimed to address their concerns, and we would appreciate further engagement with our new replies. Please let us know if there are any other issues before the review deadline of Nov 26th 11:59pm AOE.

---

> > > ### Comment · Reviewer_es5D · 2024-11-26
> > >
> > > I appreciate you addressing some of my concerns. Given I had limited concerns and some of the suggestions are outside the scope of this paper, I am keeping my score as is. I want to once again congratulate the authors on what I believe to be a strong paper.

---

> > > > ### Author Response · Authors · 2024-11-30
> > > >
> > > > We thank the reviewer for their congratulations and for their helpful comments during this review process.

---

### Official Review · Reviewer_1oHR · 2024-11-02

**Soundness:** 2
**Presentation:** 2
**Contribution:** 3
**Rating:** 5
**Confidence:** 2

**Summary:**

The paper investigated whether SAEs trained on different SAEs learn similar features. They apply techniques that try to both match SAE features directly, and also techniques which find similar subspaces. The paper compares an SAEs trained on Pythia 70m to SAEs trained on Pythia 160m, as well as comparing Gemma-2b SAEs to a Gemma-2-2b SAEs. The authors find evidence that at middle layers of the model SAEs learn similar representations.

**Strengths:**

The techniques the paper uses, SVCAA and RSA, seem to be novel when applied to SAE features. The paper compares their results to a baseline of randomly pairing up features and subspaces. Comparing SAEs from different models is also not well studied and is a good domain to explore.

**Weaknesses:**

The paper claims to find universal representations across SAEs from a wide variety of LLMs, but only looks at SAEs within the same LLM family (Pythia to Pythia, and Gemma to Gemma) which likely share a similar architecture and training corpus. It is also not clear what specific SAEs are used aside from saying they are loaded using SAELens. It is likely that the Gemma-2-2b SAEs are from Gemma Scope, but Gemma Scope is not cited and Gemma Scope has a large number of SAEs for every layer of the model with different L0 values. The techniques only seem to offer statistically significant improvements over randomly selecting SAE features in the middle layers of the model. It is unclear if the techniques failing to work in early and later layers of the model is evidence of problems with the technique or evidence that the SAEs are learning different representations. It seems surprising to me that very similar LLMs like Gemma 1 and Gemma 2 have completely unrelated representations of concepts except in middle layers.

The case the paper makes would be bolstered by comparing different SAEs trained on the same model (e.g. the multiple SAEs per layer in Gemma Scope), or multiple SAEs trained with different seeds, to rule out issues with the technique itself. The claim of universality also requires comparing SAEs across model families, not just within a model family.

**Questions:**

- In the paper, it says "features may be similar relation-wise across spaces, but not rotation-wise due to differing basis". What does it mean to be similar relation-wise? Is this referring to matching feature neurons directly?
- "layer 0 contains few discernible, meaningful, and comparable features" - why would layer 0 contain few meaningful features? layer 0 is the token embeddings, shouldn't embeddings contain meaningful features?
- In figures 2 and 3, does a score of 0.2 for SVCCA or RSA mean that only 20% of feature subspaces match across SAEs according to the metric?
- Why do RSA and SVCCA give such different values for similarity for the same model and same layers?
- In section 4.3, it says "We find that for all layers and for all concept categories, Test 2 described in §3 is passed. Thus, we only report specific results for Test 1 in Tables 1 and 2". Does this mean test 2 failed? I'm confused what this means.
- Section 4.3 mentions non-semantic subspaces. What are non-semantic subspaces? How do you know if a subspace is semantic or non-semantic?

---

> ### Author Response · Authors · 2024-11-21
> **Response to Reviewer 1oHR**
>
> We thank the reviewer for their comments and will address their points in our response.
>
> > only looks at SAEs within the same LLM family (Pythia to Pythia, and Gemma to Gemma) which likely share a similar architecture and training corpus.
>
> Experiments on cross-model families require a new approach due to the individual SAE feature pairings utilizing activation correlation by token, which requires each model to use the same tokenizer. At this moment, we are currently working on experimenting with possible solutions to this problem.
>
> Though this paper lacks cross-model family experiments, the experiments in the paper take the first steps in comparing SAEs across models with more than one layer, and which are LLMs that are not toy models. Thus, as a standalone, we believe it is novel enough, and follow-up papers in the future can conduct experiments with cross-model families.
>
> > Gemma Scope is not cited
>
> We cited Gemmascope on line 248.
>
> > It is also not clear what specific SAEs are used aside from saying they are loaded using SAELens. It is likely that the Gemma-2-2b SAEs are from Gemma Scope, but Gemma Scope is not cited and Gemma Scope has a large number of SAEs for every layer of the model with different L0 values.
>
> Gemma scope provides a ‘canonical SAE’ that is maintained through SAELens. The canonical SAEs are those with L0 value closest to 100 out of all the SAEs provided. Links to these SAEs are given as follows:
> Gemma-1-2b with specified L0 values:
> https://github.com/jbloomAus/SAELens/blob/a470460/sae_lens/pretrained_saes.yaml#L403
>
> Gemma-2-2b:
> https://github.com/jbloomAus/SAELens/blob/a470460/sae_lens/pretrained_saes.yaml#L1667
> We will add an additional Appendix section and enumerate these values explicitly.
>
> >  It is unclear if the techniques failing to work in early and later layers of the model is evidence of problems with the technique or evidence that the SAEs are learning different representations.
>
> Based on previous work, it is expected that the middle layers would contain more meaningful, and thus likely more similar, representations. Previous work such as Rimsky et al. (2023) and Mini et al. (2023) [1, 2] have shown that steering in the middle layers yielded the most notable changes on model behavior.
>
> [1]: https://arxiv.org/abs/2312.06681
> [2]: https://arxiv.org/abs/2310.08043
>
> > The case the paper makes would be bolstered by comparing different SAEs trained on the same model (e.g. the multiple SAEs per layer in Gemma Scope)
>
> We are currently working on this to include in an updated version of the paper.
>
> > In the paper, it says "features may be similar relation-wise across spaces, but not rotation-wise due to differing basis". What does it mean to be similar relation-wise?
>
> Relation-wise means feature distances (eg. king - queen), so being more similar relation-wise means the pairwise distances for all feature pairs is more similar.
>
> > "layer 0 contains few discernible, meaningful, and comparable features" - why would layer 0 contain few meaningful features? layer 0 is the token embeddings, shouldn't embeddings contain meaningful features?
>
> Our intuition is that early layers, especially very early ones, do not handle much semantic processing at higher levels. Rather, they may be concerned with more syntactic computations. In particular Layer 0 has embeddings of input tokens, but not features extracted from those token embeddings. In this paper, we aim to compare universality of higher level features and concepts that can be disentangled by SAEs, rather than lower level features.
>
> > In figures 2 and 3, does a score of 0.2 for SVCCA or RSA mean that only 20% of feature subspaces match across SAEs according to the metric?
>
> 0.2 is the SVCCA or RSA similarity score, not a proportion.
>
> > Why do RSA and SVCCA give such different values for similarity for the same model and same layers?
>
> RSA and SVCCA measure different properties. RSA measures pairwise differences, and SVCCA measures alignment of spaces.
>
> > In section 4.3, it says "We find that for all layers and for all concept categories, Test 2 described in §3 is passed. Thus, we only report specific results for Test 1 in Tables 1 and 2". Does this mean test 2 failed?
>
> This means that experiments for all (layer, concept category) combinations passed Test 2.
>
> > Section 4.3 mentions non-semantic subspaces. What are non-semantic subspaces? How do you know if a subspace is semantic or non-semantic?
>
> In lines 203 to 206, Test 2 implicitly defines non-semantic subspaces as “randomly selected feature subsets of the same sizes as the semantic subspaces”. This means the features in this subspace are not expected to be grouped under a unifying semantic concept.

---

> > ### Comment · Reviewer_1oHR · 2024-11-24
> >
> > Thank you for clarifying the questions. I've updated my score.

---

> > > ### Author Response · Authors · 2024-11-25
> > > **Response to reviewer 1oHR**
> > >
> > > We are glad to have clarified your questions and we would be happy to answer any further questions you might have.
> > >
> > > If we have answered all of your concerns - we kindly ask you to increase your score such that it could result in the acceptance of our paper.
> > >
> > > Thank you.

---

> > > > ### Author Response · Authors · 2024-11-25
> > > >
> > > > Additionally, we thank the reviewer for their updated score and are happy to address any other issues the reviewer may still have that may lead to a further increase in score.

---

### Official Review · Reviewer_4nyz · 2024-11-03

**Soundness:** 3
**Presentation:** 3
**Contribution:** 2
**Rating:** 5
**Confidence:** 4

**Summary:**

The paper conducts exploratory experiments to measure representational similarity between the overcomplete bases found by sparse autoencoders (SAEs) across different LLMs and layers therein.

Specifically:
- suppose we are given two SAEs trained on residual stream activations (possibly coming from different models and/or layers within the model) which have the same number $d_{hidden}$ of hidden latents
- consider the decoder matrices $D_1,D_2$ of these SAEs, which hold the $d_{hidden}$ vectors in the overcomplete basis the SAE uses to sparsely code residual stream activations
- pair up the SAE latents: given a latent in one SAE, pair it up with the latent from the other SAE that has the highest correlation of activations over some set of texts (there are some technical details on how to clean up this mapping to make it injective);
- after this pairing we have two "aligned" matrices of latents $D_1',D_2'$, to which we apply two representation similarity methods:
	- SVCCA (Singular value canonical correlation analysis), which first takes the left singular vectors of $D_1', D_2'$, and then runs canonical correlation analysis on these two sets of vectors, iteratively finding linear combinations of vectors on either side that maximize correlation (equivalently dot product) and removing the pair. The final measure of correlation is the average of the correlations of all pairs;
	- RSA (Representational similarity analysis): here we first build a distance matrix over the vectors in $D_1'$ (and similarly $D_2'$), where the $ij$ entry is the Euclidean distance between the $i$-th and $j$-th latent vectors. Then, the similarity of the two resulting distance matrices is evaluated via Spearman (rank) correlation.
- beyond evaluating the correlations as above, their statistical significance is measured via p-values against a random pairing of the original SAE latents $D_1,D_2$ designed to simulate a null condition.
- these experiments are ran on pairs of SAEs trained on different Pythia models (70M and 160M) and their layers, as well as pairs of SAEs trained on different Gemma-2B model versions (1 and 2) and their layers.

The authors report some findings, such as achieving statistical significance under their null test for most pairs of SAEs, and finding the middle-layer SAEs are most similar.

**Strengths:**

- The paper studies an underexplored problem, the extent to which SAE latents are universal across models
- The paper proposes some interesting metrics for SAE similarity, such as representational similarity analysis (RSA)
- The result that pairs of middle-layer SAEs are most similar is mildly interesting

**Weaknesses:**

- The results are exploratory, and there is no strong takeaway for the interpretability research community.
	- The overall conclusion seems to be that "SAE latents are kind of similar between layers/models", but it is difficult to contextualize the magnitude of the effect. It would be valuable if there is a comparison to similarity in the neuron basis. Currently, it is unclear what is gained by asking the question of SAE latents as opposed to original neurons in the model.
	- It is unclear whether comparing SAE latents across different layers is an interesting question; the main motivation for studying universality is to assure us that training SAEs on a different model using a different dataset sample will discover "the same" concepts *in aggregate*. In particular, the current consensus in the field is that early, middle and late model layers all have distinct roles in computation and likely represent distinct concepts (also see some recent work for why trying to align SAE latents in faraway layers "naively" may be the wrong approach https://transformer-circuits.pub/2024/crosscoders/index.html)
- The use of p-values to quantify the significance of the results is not very illuminating, because even small effect sizes can often achieve small p-values.
	- In particular, the results on low p-values of SVCCA are perplexing. SVCCA is permutation-invariant, so whether or not features were paired together beforehand (vs randomly paired as in the null condition) should not matter for the final result. I suspect the low SVCCA p-values may be due to the post-processing step after feature pairing that prunes the mapping to make it better behaved.
	- Regardless of this issue, it is altogether not surprising the p-values are low for the other method (RSA).
- The Methodology section is not very clearly written

**Questions:**

- Have you looked at the latents that get paired between SAEs in the same (or similar) layers using your approach based on activation correlations? Do they seem to be interpretable? Do they have similar interpretations?
- Why should we be interested in measuring the similarity between SAEs trained on distant model layers? In particular, when you look at the paired features across distant layers, do they seem to have similar interpretations?
- What baseline similarity should we expect between different SAEs random seeds for the same training set of activations, the same model and the same layer? Similarly, what should we expect when any of these are made different instead of the same?

---

> ### Author Response · Authors · 2024-11-21
> **Response to Reviewer 4nyz**
>
> We thank the reviewer for their comments and will address their points in our response.
>
> > The overall conclusion seems to be that "SAE latents are kind of similar between layers/models", but it is difficult to contextualize the magnitude of the effect. It would be valuable if there is a comparison to similarity in the neuron basis. Currently, it is unclear what is gained by asking the question of SAE latents as opposed to original neurons in the model.
>
> We are currently working on these experiments to include in an updated version of the paper.
>
> > It is unclear whether comparing SAE latents across different layers is an interesting question; the main motivation for studying universality is to assure us that training SAEs on a different model using a different dataset sample will discover "the same" concepts in aggregate. In particular, the current consensus in the field is that early, middle and late model layers all have distinct roles in computation and likely represent distinct concepts (also see some recent work for why trying to align SAE latents in faraway layers "naively" may be the wrong approach
>
> To clarify, we would like the reviewer to expand more on what “in aggregate” means so we can better address this concern. Is this phrase referring to concepts across layers?
>
> > SVCCA is permutation-invariant, so whether or not features were paired together beforehand (vs randomly paired as in the null condition) should not matter for the final result.
>
> As discussed in Klabunde et al. (2023) [1], the permutation invariances refers to the columns of the representation matrices, rather than the rows:
> "Permutations (PT). A similarity measure m is invariant to permutations if swapping columns of the representation matrices R, that is, reordering neurons, does not affect the resulting similarity score."
>
> Thus, the row pairing can still affect the score.
>
> [1] https://arxiv.org/pdf/2305.06329
>
> > I suspect the low SVCCA p-values may be due to the post-processing step after feature pairing that prunes the mapping to make it better behaved.
>
> Even if we don’t post-process the data, we find the experiments still have low p-values for SVCCA.
>
> > The Methodology section is not very clearly written
>
> We thank the reviewer for these concerns but would like the reviewer to point out more specific areas in this writing that are not clear.
>
> > Have you looked at the latents that get paired between SAEs in the same (or similar) layers using your approach based on activation correlations? Do they seem to be interpretable? Do they have similar interpretations?
>
> Yes, we have found that they are interpretable and have similar interpretations. We can include examples of these in the Appendix of an updated version of the paper.
>
> > Why should we be interested in measuring the similarity between SAEs trained on distant model layers? In particular, when you look at the paired features across distant layers, do they seem to have similar interpretations?
>
> In the field of evaluating model similarity,  previous papers like Kornblith et al. (2019) [1] measured similarity across every layer pair, including distant ones. This provides a comprehensive overview. Additionally, comparing distant  layers allows us to show that middle layers have the most similarity. We have also found that paired features across distant layers have similar interpretations, though we have not explicitly measured this yet.
> [1]: https://arxiv.org/abs/1905.00414
>
> > What baseline similarity should we expect between different SAEs random seeds for the same training set of activations, the same model and the same layer? Similarly, what should we expect when any of these are made different instead of the same?
>
> Bricken et al. (2023) [1] already tested different SAEs trained with random seeds on 1 layer toy models, so we chose to focus on pretrained SAEs “in the wild” that have different number of layers. We expect these SAEs, as they are more similar to one another than the pretrained SAEs we analyzed, would already be highly similar. Most likely, models with greater architectural differences would differ more greatly.
> [1] https://transformer-circuits.pub/2023/monosemantic-features/index.html

---

> > ### Author Response · Authors · 2024-11-25
> >
> > We thank the reviewer again for their review. We have posted new responses that have aimed to address their concerns, and we would appreciate further engagement with our new replies. Please let us know if there are any other issues before the review deadline of Nov 26th 11:59pm AOE.

---

> > > ### Comment · Reviewer_4nyz · 2024-11-27
> > >
> > > Thank you for your thoughtful response.
> > > - Regarding SVCCA: you are absolutely correct, and I was wrong about this - indeed, the order of rows matters. I no longer find the results puzzling. Thank you for correcting me, and I apologize for my confusion
> > > - by "in aggregate" I mean that we're interested in the ensemble of concepts learned by the model, and how they are causally linked. For example, see the recent work on crosscoders by Anthropic https://transformer-circuits.pub/2024/crosscoders/index.html for an interesting take on this question.
> > > - regarding the clarity of the methods section - I eventually understood what's going on, but had to go back and forth several times. Here are some notes:
> > >     - the presentations feels split across many parts. For example, there are two paragraphs titled "Assessing Scores with Baseline."
> > >     - the motivation for pairing neuron weights is unclear
> > >     - you keep mentioning the "permutation issue" and "rotational issue". These names are confusing, as they are a very roundabout way of referring to what is actually meant, which from what I understand is either a step in your evaluation algorithm or a set of metrics associated with that step.
> > >     - it is also confusing how these concepts relate to one another. From what I understand, the "rotational issue" stuff is applied *after* the "permutation issue" stuff is applied. But it is not that clear from the high-level description in the methods section (the first `\enumerate` environment) that the output of one is used as input to the next.
> > >     - the section on "Metrics for the permutation issue" combines general definitions (the mean activation correlation) with ad-hoc experimental procedure applied for this paper's experiments ("filter by top tokens", etc). Ideally, these things should be more clearly delineated.
> > >
> > > However, my main concern with this paper is the limited motivation and novelty, which has not been addressed during the discussion process. Therefore I will not adjust my score at present.

---

> > > > ### Author Response · Authors · 2024-11-30
> > > >
> > > > We thank the reviewer for their helpful writing suggestions. We will describe how we will improve each point in an updated version. We can also upload this updated version as a pdf during this review period:
> > > >
> > > > > the presentations feels split across many parts. For example, there are two paragraphs titled "Assessing Scores with Baseline."
> > > >
> > > > Each of the subsections titled "Assessing Scores with Baseline” was a subsection for two separate experiment types that each used different baselines. More specifically, the first subsection is a general approach to carrying out the experiments by comparing the results to random pairings, while the second subsection discusses tests specific to only semantic subspace experiments.
> > > >
> > > > To make these paragraphs clearer, we can rename these terms to be more accurate and specific: the first can be renamed as “Steps for Feature Space Similarity Experiments” and the second can become “Baselines for Semantic Subspace Matching Experiments”.
> > > >
> > > > > the motivation for pairing neuron weights is unclear
> > > >
> > > > To compare feature spaces, the individual features (represented as SAE neurons) should first be paired because the representational space measures used require individual data points (the rows of a matrix) in each space to be the same. In previous papers like Kornblith et al. (2019) [1], because activation spaces were compared, the data points used in both models were the same inputs. In our paper, to compare feature weight spaces, the data points used in both models were the same features (or SAE neurons).
> > > >
> > > > [1]: https://arxiv.org/abs/1905.00414
> > > >
> > > > For instance, as described in lines 132 to 140, RSA computes, for each space, an RDM that captures the pairwise distances between all possible data points. Thus, model A’s RDM would contain information about the distance between the feature pair (King, Queen), and so would model B’s RDM. However, the question is: which would be the feature considered to activate on “king” in model A, and which would be its analogous feature that activates on “king” in model B? This individual data point mapping is thus determined via mean activation correlation. There may be multiple features activating on “king” in each model, so the approach attempts to find the best individual match.
> > > >
> > > > > you keep mentioning the "permutation issue" and "rotational issue". These names are confusing, as they are a very roundabout way of referring to what is actually meant, which from what I understand is either a step in your evaluation algorithm or a set of metrics associated with that step.
> > > >
> > > > We can rewrite the paper to first more clearly define what the “permutation issue” and the “rotational issue” are before using these terms. We described above what the permutation issue is; in an updated version of the paper, we can explicitly state these definitions (and put each phrase in bold text) as:
> > > >
> > > > Permutation Issue: This problem involves finding a mapping of individual features across models.
> > > >
> > > > Rotational Issue: This problem involves measuring global similarity after an individual feature mapping. Two spaces may not be rotationally aligned because the bases of each space are likely different from one another. More specifically, we employ two different measures to measure global similarity in two ways: 1) Using SVCCA to determine how well subspaces align, and 2) Using RSA to determine the similarity of all feature relations like king-queen.
> > > >
> > > > We can also rename each term to be more specific and less ambiguous. The first problem can be renamed as the “Feature Permutation (or Pairing) Problem” that involves finding which permutation (or mapping) is better to use, and the second problem can be renamed as the “Rotational Feature Space Problem” that involves aligning two different bases.
> > > >
> > > > > it is also confusing how these concepts relate to one another. From what I understand, the "rotational issue" stuff is applied after the "permutation issue" stuff is applied. But it is not that clear from the high-level description in the methods section (the first \enumerate environment) that the output of one is used as input to the next.
> > > >
> > > > In lines 178 to 184, we can state the inputs and outputs of each step of this algorithmic procedure clearer as follows, matching the steps in Figure 8:
> > > >
> > > > Step 1- Input: two SAE decoder weight matrices (with unpaired rows)
> > > >
> > > > Step 1- Output: A list of pairings of each feature A with its max activation correlated feature from model B.
> > > >
> > > > Step 2- Input: the list of pairings from Step 1.
> > > >
> > > > Step 2- Output: A row-wise rearrangement of the order of features in model B’s matrix based on the list of pairings.
> > > >
> > > > Step 3- Input: the rearranged matrices from Step 2.
> > > >
> > > > Step 3- Output: A Paired Score for each representational similarity measure

---

> > > > > ### Author Response · Authors · 2024-11-30
> > > > >
> > > > > > the section on "Metrics for the permutation issue" combines general definitions (the mean activation correlation) with ad-hoc experimental procedure applied for this paper's experiments ("filter by top tokens", etc). Ideally, these things should be more clearly delineated.
> > > > >
> > > > > We can more specifically delineate general definitions vs more ad-hoc terms by putting the ad-hoc terms in quotes or using different fonts (such as italics) to distinguish them. A section of the general definitions with citations can also be included in the Appendix. We are also open to other more specific suggestions the reviewer may have for this point.
> > > > >
> > > > > > However, my main concern with this paper is the limited motivation and novelty, which has not been addressed during the discussion process. Therefore I will not adjust my score at present.
> > > > >
> > > > > We described the motivation and novelty of this paper in both the Introduction (Section 1) and Related Works (Section 5). To make these points clearer, we will look into moving parts of the Related Work to the Introduction. We address these concerns as follows:
> > > > >
> > > > > Motivation: In the introduction, we discuss how SAEs have recently shown good results for disentangling polysemantic representations in LLMs. As SAEs, in the context of tackling this issue, are a new tool in the field of interpretability, it is important to determine if their results are transferable from tested models to other models; this allows their results to be potentially be more generalizable, so that we may expect studies on a few examples to also be replicable in other models. Additionally, measuring universality is important for transferable features and feature relations, which may reduce redundant SAE training by making training more efficient via feature reuse.
> > > > >
> > > > > Novelty: In Lines 405 to 407, we state:
> > > > >
> > > > > “To the best of our knowledge, only Bricken et al. (2023) [2]  has done a quantitative study on individual SAE feature similarity for two 1-layer toy models, finding that individual SAE feature correlations are stronger than individual LLM neuron correlations; however, this study did not analyze the global properties of feature spaces.”
> > > > >
> > > > > Thus, our paper is the first to measure SAE feature space similarity, not just individual feature similarity, across models. It is also the first to look at “real world” models used in practice, and not just on a pair of toy models.
> > > > >
> > > > > [2] https://transformer-circuits.pub/2023/monosemantic-features/index.html
> > > > >
> > > > > > by "in aggregate" I mean that we're interested in the ensemble of concepts learned by the model, and how they are causally linked. For example, see the recent work on crosscoders by Anthropic
> > > > >
> > > > > The causal links between an ensemble of concepts is an interesting topic that we are looking to investigate further.
> > > > >
> > > > > One note is that our paper does not follow up on the recent work on crosscoders by Anthropic [3] as it was released after our paper was submitted to ICLR. Thus, we can investigate this in future work.
> > > > >
> > > > > [3] https://transformer-circuits.pub/2024/crosscoders/index.html

---

### Meta-Review · Area_Chair_g5sB · 2024-12-22

**Metareview:**

This paper investigates feature similarity between SAEs trained on different models and different layers, using techniques such as SVCAA and RSA. It finds empirical evidence supporting the existence of universal features across different LLMs. Reviewers have mixed opinions about this paper. While they generally acknowledge the importance, novelty, and clear methodology for comparing SAE spaces developed in the paper, they also raise concerns about the baseline choice, the limitation to comparison within the same model family, and the limited experiment scope. The AC agrees that the paper could be further strengthened and made more convincing by removing some of these limitations.

**Additional Comments On Reviewer Discussion:**

The discussion phase resolved some confusion in the technical aspects of the work. A shared concern that remained among reviewers is the limitation to within-model-family comparison, which significantly limits the contributions of this paper. The authors were upfront about this limitation.

---

### Decision · Program_Chairs · 2025-01-22

Reject